# Printable magnesium ion quasi-solid-state asymmetric supercapacitors for flexible solar-charging integrated units

Zhengnan Tian[1,6], Xiaoling Tong[1,6], Guan Sheng[2], Yuanlong Shao[2]*, Lianghao Yu[1], Vincent Tung[2], Jingyu Sun[1,3]*, Richard B. Kaner[4]* & Zhongfan Liu [1,3,5]

Wearable and portable self-powered units have stimulated considerable attention in both the scientific and technological realms. However, their innovative development is still limited by inefficient bulky connections between functional modules, incompatible energy storage systems with poor cycling stability, and real safety concerns. Herein, we demonstrate a flexible solar-charging integrated unit based on the design of printed magnesium ion aqueous asymmetric supercapacitors. This power unit exhibits excellent mechanical robustness, high photo-charging cycling stability (98.7% capacitance retention after 100 cycles), excellent overall energy conversion and storage efficiency ($\eta_{overall} = 17.57\%$), and outstanding input current tolerance. In addition, the Mg ion quasi-solid-state asymmetric supercapacitors show high energy density up to 13.1 mWh cm$^{-3}$ via pseudocapacitive ion storage as investigated by an operando X-ray diffraction technique. The findings pave a practical route toward the design of future self-powered systems affording favorable safety, long life, and high energy.

[1] College of Energy, Soochow Institute for Energy and Materials InnovationS (SIEMIS), Jiangsu Provincial Key Laboratory for Advanced Carbon Materials and Wearable Energy Technologies, Soochow University, 215006 Suzhou, P. R. China. [2] College Physical Sciences and Engineering Division, King Abdullah University of Science and Technology, Thuwal 23955-6900, Saudi Arabia. [3] Beijing Graphene Institute (BGI), 100095 Beijing, P. R. China. [4] Department of Chemistry and Biochemistry, Department of Materials Science and Engineering, and California NanoSystems Institute, University of California, Los Angeles (UCLA), Los Angeles, CA, USA. [5] Center for Nanochemistry (CNC), College of Chemistry and Molecular Engineering, Peking University, 100871 Beijing, P. R. China. [6]These authors contributed equally: Zhengnan Tian, Xiaoling Tong. *email: yuanlong.shao@gmail.com; sunjy86@suda.edu.cn; kaner@chem.ucla.edu

The growing demand for miniaturization and multi-functionalization of portable electronics has spurred the development of integrated energy systems that can hybridize energy harvesting and storage in a flexible and even wearable fashion[1]. In terms of energy-harvesting modules, photovoltaic (PV) devices, especially silicon solar cells, have become ubiquitous for harnessing clean and accessible solar energy. However, due to the intermittent and unpredictable nature of sunlight, the fluctuating output energy of PV panels is highly unlikely to support the direct powering of wearable electronics; thus, energy needs to be stored in a reliable module for further use on demand. Among a plethora of emerging energy storage technologies, supercapacitors offer high power density and excellent tolerance to the variation of input current, thereby being perfect for accommodating PV modules for building efficient self-powered platforms[2]. Despite these meritorious properties, supercapacitors suffer from relatively low energy density when compared with metal-ion battery systems, which significantly impedes their applications in the realm of wearable technology. In this regard, developing asymmetric supercapacitors (ASCs) remains a feasible solution: their extended operating voltage favors enhanced energy density without sacrificing high power capability and good current tolerance; hence, benefiting the overall performance of PV-supercapacitor integrated systems[3]. Recently, Zhang et al.[4] demonstrated a flexible photo-capacitor based on a $Co_9S_8$-$MnO_2$//$TiO_2$ asymmetric electrode design. In our previous work, a wearable photo-rechargeable lithium-ion hybrid capacitor (LIC) also demonstrated a high overall efficiency up to 8.41%[5].

Safety is the primary concern for wearable energy systems, which therefore demand aqueous electrolytes along with non-flammable electrodes. Although it is difficult for an aqueous electrolyte system to achieve a high working voltage compared to organic electrolytes (2.5–4.0 V), the two order of magnitude greater ionic conductivity endows it with fast ion transport and hence, high-rate performance. Furthermore, it is interesting to note that neutral aqueous electrolyte-based ASCs can achieve a wide working voltage from ~1.6 to 2.0 V, which exceeds the theoretical limit of the water splitting potential, 1.23 V. Such a large operating voltage is achieved by high hydrogen and oxygen evolution overpotentials in a neutral aqueous system. Of particular note, these neutral electrolytes readily overcome the hazardous concerns of flammability and corrosion that occur with organic and acidic/alkaline electrolytes; thus, overcoming the substantial safety issues encountered with wearable devices. As such, exploiting neutral electrolytes with multivalent metal-ions ($Mg^{2+}$, $Al^{3+}$, $Zn^{2+}$, $Ca^{2+}$, etc.) for aqueous ASCs represents a promising strategy to synchronously address the safety challenges while boosting energy density[6,7]. However, until now, the development of a wearable solar-charging power unit employing multivalent metal-ion ASCs based on directly printing designs is still in its nascent stage, and hence in need of systematic exploration.

Herein, we devise aqueous Mg ion ASCs targeting the construction of wearable solar-charging integrated power units. Such flexible, quasi-solid-state ASCs ($MnO_2$//VN) are assembled with nanoflower-like $MnO_2$ as the positive electrode, nanowire-shaped VN as the negative electrode, and $MgSO_4$-PAM gel as the electrolyte, synergistically achieving a wide potential window (up to 2.2 V), favorable volumetric energy density (up to 13.1 mWh cm$^{-3}$), high power density (up to 440 mW cm$^{-3}$), and outstanding cycling stability (capacitance retention up to 95% after 5000 cycles). The Mg ion electrochemical system developed in this work has the following attributes: (i) a superior charge storage efficiency due to the dual-electron transfer possible with $Mg^{2+}$-ion intercalation in comparison with typical alkali metal-ion ($Li^+$, $Na^+$, $K^+$) systems; (ii) a facile redox process enabled by the highly efficient surface

ion migration and near-surface ion intercalation of nano-sized $MnO_2$ and VN; and (iii) a wider voltage window owing to the negative hydrogen evolution reaction overpotential on VN surfaces. The Mg ion transport mechanism at the VN anode is further probed using operando X-ray diffraction (XRD) and ex situ X-ray photoelectron spectroscopy (XPS) techniques. Moreover, direct printing of such device arrays on the rear of flexible solar cells enables the fabrication of solar-charging power units with simplified configurations and mechanical robustness. Importantly, the printable and flexible integrated system thus-derived shows remarkable mechanical robustness, excellent photo-charging cycling stability (98.7% capacitance retention after 100 cycles), and efficient powering of electronic devices (e.g. a smart watch and a light-emitting diode), demonstrating great potential as self-powered wearable energy sources for safe use in practical scenarios.

## Results

**Illustrating the configuration of solar-charging units.** Figure 1a (i) illustrates the configuration of a flexible solar-charging self-powered unit, encompassing an energy-harvesting module (i.e. a flexible solar cell), an energy storage module (i.e. a printed quasi-solid-state ASC array on a polyimide substrate), and a plastic film capping layer. Note in Fig. 1a (ii) that the quasi-solid-state ASC is assembled by using vanadium nitride (VN) as the negative electrode, manganese oxide ($MnO_2$) as the positive electrode, and $MgSO_4$-polyacrylamide (PAM) gel as the electrolyte. Such an aqueous supercapacitor device is operated based on the pseudo-capacitive intercalation/de-intercalation of $Mg^{2+}$ ions between VN and $MnO_2$ electrodes. For the integrated unit, upon exposure to sunlight, the solar cell component enables the conversion of photo-irradiation into electricity and charges the supercapacitor simultaneously (photo-charging). The photo-charged super-capacitor can provide stored power for electronic devices anytime on demand (discharging). In addition, the entire integrated system complies with a thin-film planar design, which endows it with good flexibility. As a proof-of-concept demonstration, a solar-charging self-powered unit can be directly worn and function as a reliable power source for powering a portable electronic watch, as shown in Fig. 1a (iii).

**Characterizing as-prepared VN nanowires.** The synthetic route to VN nanowires was adopted from a reported recipe (Supplementary Fig. 1a)[8]. Scanning electron microscopy (SEM; Fig. 1b) images show the typically porous nanowire structure of VN, with uniform 120–150 nm in width and ~2 μm in length. Transmission electron microscopy (TEM; Fig. 1c) presents the detailed morphology of the porous VN nanowires, showing many interconnected open pores. Elemental mapping of the as-prepared VN reveals a homogeneous distribution of nitrogen and vanadium (Supplementary Fig. 1b). The appearance of an oxygen signal suggests the existence of surface-bound oxygen upon exposure to air. Further XPS analysis indicates the dominance of the $V^{3+}$-binding state as well as the V−N signal (Supplementary Fig. 1c, d), implying the successful fabrication of VN. Note that there are no XPS peaks indicative of $V^{4+}$ and $V^{5+}$ states, which can be attributed to the excessive reduction from the long-term and high-temperature $NH_3$ annealing process (600 °C for 3 h). Aberration-corrected scanning transmission electron microscopy (STEM) observations display the lattice fringes of VN (Fig. 1c inset; Supplementary Fig. 1e–g), with a lattice spacing of 0.21 nm matching well with the VN (200) plane. Figure 1d depicts the X-ray diffraction (XRD) pattern of synthesized VN samples. Characteristic peaks are found at 37.9° (111), 44.0° (200), and 64.1° (220), verifying the well-defined crystal structure of VN

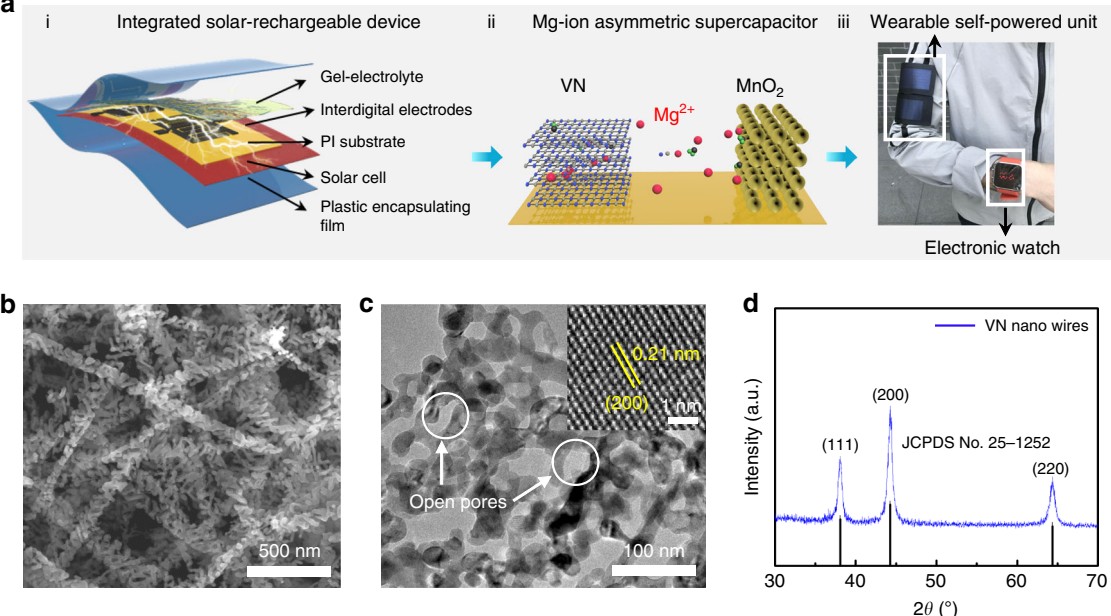

**Fig. 1** Schematic illustration of the solar-charging integrated unit and characterization of VN nanowires. **a** Configuration of the solar-charging self-powered unit and proof-of-concept demonstration of wearable applications. **b** SEM image of as-prepared porous VN nanowires. **c** TEM image of VN nanowires. Inset: High-resolution STEM image. **d** XRD pattern of VN

(JCPDS No. 25-1252). It is anticipated that such nanowire-shaped VN is key to achieving favorable electrochemical performances when used as electrodes due to: (1) the large aspect ratio, intrinsic porous structure, and nanometer sized dimensions (<20 nm) of the VN nanowires which endow them with high-rate capabilities and near-surface $Mg^{2+}$ intercalation pseudo-capacitive behavior, a crucial factor for their superior rate capability and cycling stability. (2) the nanowire-shaped microstructure could become interwoven with each other thereby avoiding aggregation upon cycling, which has often been seen when using nanoparticle or nanosheet morphologies.

**Investigating electrochemical properties of VN nanowires**. The electrochemical performance of VN was systematically examined via a three-electrode cell system. Cyclic voltammetry (CV) measurements at a scan rate of 10 mV s$^{-1}$ were carried out in various neutral aqueous electrolytes affording identical cation concentrations (Fig. 2a), including 0.5 M $Li_2SO_4$, 0.5 M $Na_2SO_4$, 0.5 M $K_2SO_4$, and 1.0 M $MgSO_4$. It is interesting to note that the VN electrode exhibits the largest CV area in the Mg ion electrolyte. Figure 2b summarizes the gravimetric-specific capacitances calculated based on the CV profiles. The VN electrode manifests a specific capacitance of 230 F g$^{-1}$ in the $MgSO_4$ electrolyte, far superior to those achieved in $Li_2SO_4$ (120 F g$^{-1}$), $Na_2SO_4$ (100 F g$^{-1}$), and $K_2SO_4$ (70 F g$^{-1}$). Such markedly enhanced capacitance in the $Mg^{2+}$ electrolyte mainly stems from the following factors: (i) the bivalent metal-ion ($Mg^{2+}$) possesses a higher theoretical capacitance due to dual-electron transfer during redox reactions. The possible Faradaic reactions of VN in an aqueous $Mg^{2+}$ electrolyte can be described as follows[9]:

$$VN_xO_y + zMg^{2+} + z2e^- \leftrightarrow VN_xO_yMg_z. \qquad (1)$$

In this respect, the capacitance stored in the $Mg^{2+}$ system should be twice that of other monovalent cation systems, which is in good agreement with the capacitance derived from the CV measurements. (ii) The relatively smaller ionic radius of $Mg^{2+}$

ions ($Mg^{2+}$: 0.72 Å, $Li^+$: 0.76 Å, $Na^+$: 1.02 Å, $K^+$: 1.51 Å)[10] leads to pseudo-capacitive-dominated charge storage behavior of the VN electrode.

Electrochemical impedance spectroscopy (EIS) studies offer further insights into the ionic and electronic transport properties of the VN electrode in different cation electrolytes (Supplementary Fig. 2a, b). At the low-frequency region, the steep shape of the straight lines reveals a typical capacitive behavior in various electrolytes. At the high-frequency region, the internal resistance ($R_s$) shows obvious differences ($K^+$: 4 Ω, $Na^+$: 5.5 Ω, $Li^+$: 7 Ω, $Mg^{2+}$: 8.5 Ω). $R_s$ normally consists of the conductive resistance of the electrolyte, as well as the intrinsic contact resistance between electrode material and current collector. Since all the Nyquist plots were acquired based on the same electrode material and electrochemical cell configuration, the difference of $R_s$ should be mainly attributed to the diverse conductive resistances of different cation electrolytes. Because of its bivalent nature and small ionic radius, $Mg^{2+}$ ions exhibit relatively strong ionic bonds. Therefore, the diffusion of $Mg^{2+}$ ions is more sluggish than that of monovalent cations. The higher resistance polarization of $Mg^{2+}$ should result in a larger voltage hysteresis, which is helpful in extending the stable working voltage of VN in the $Mg^{2+}$ electrolyte system. A Randles equivalent circuit with relative parameters is provided (Supplementary Fig. 2c; Supplementary Table 1), accompanied by the detailed discussions shown in Supplementary Note 1. Figure 2c displays the linear sweep voltammetry (LSV) polarization curves of a VN electrode in different cation electrolytes. It is evident that the $Mg^{2+}$ system manifests the highest overpotential for hydrogen evolution. Along this line, the stable working voltage of the VN electrode in an $Mg^{2+}$ electrolyte can be extended, corroborating the EIS analysis. Figure 2d summarizes the specific capacitances of VN in the $Mg^{2+}$ electrolyte that correspond to the CV profiles at different scan rates (Fig. 2d, inset). The CV shapes are well maintained upon increasing the scan rate from 1 to 200 mV s$^{-1}$, indicating the favorable rate performance of the VN electrode in the $Mg^{2+}$ system. Note that the redox peak between −0.6 and −0.7 V only displays a slight shift when increasing the scan rates, indicating

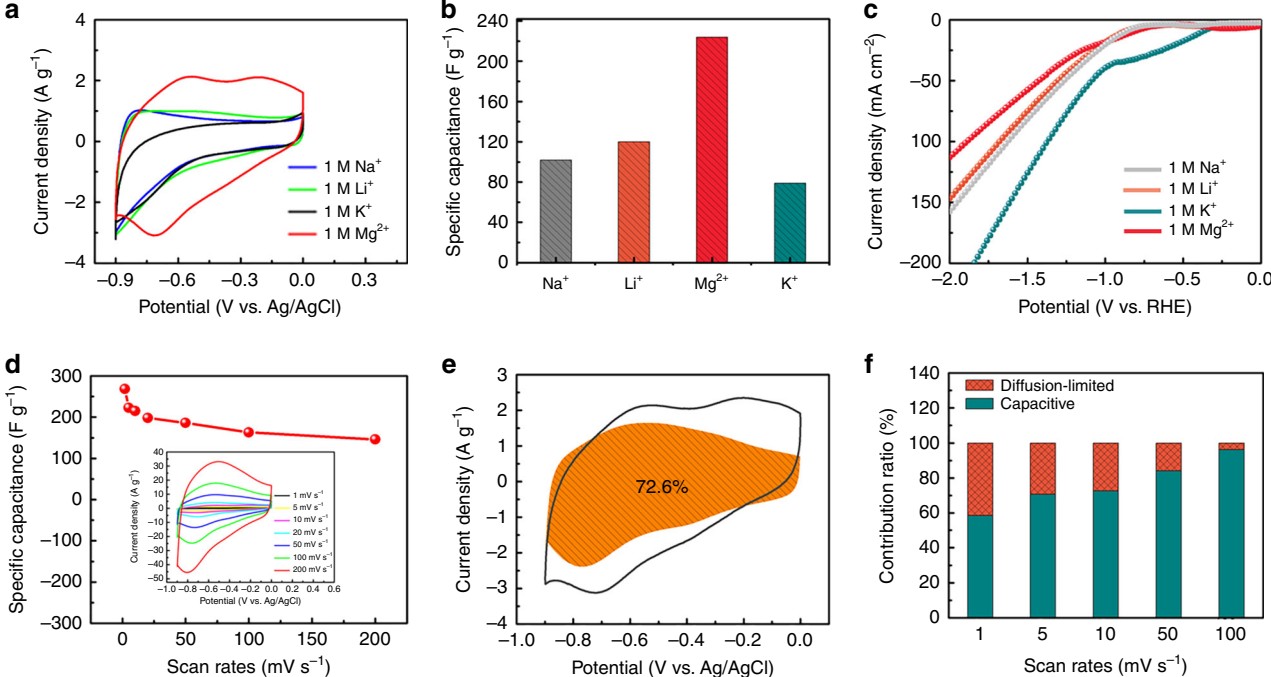

**Fig. 2** Electrochemical characterization of VN in a three-electrode configuration. **a** CV curves measured in different cation-based electrolytes at the same scan rate of 10 mV s$^{-1}$. **b** Corresponding gravimetric specific capacitances calculated from **a** in different electrolytes. **c** Polarization curves of a VN electrode in different cation-based neutral electrolytes. **d** Specific capacitances of VN in 1.0 M MgSO$_4$ electrolyte at different scan rates. Inset: Corresponding CV curves at different scan rates. **e** Capacitive and diffusion-controlled contributions to the total charge storage of VN in 1.0 M MgSO$_4$ at 10 mV s$^{-1}$. **f** Normalized contribution ratio of capacitive (cyan) and diffusion-controlled (orange) capacities at different scan rates

facile reaction kinetics during charge storage. As such, the charge stored by the Faradaic diffusion-controlled or non-Faradaic capacitive-tailored contributions can be analyzed by a *b*-value, which can be derived based on the equation $i = av^b$ [11]. In this sense, the *b*-value is respectively calculated at 0.91 and 0.90 for the anodic and cathodic peak (Supplementary Fig. 2d), implying that the contribution from capacitive behavior dominates. To further identify the contribution ratio, the current response can be expressed by combining the two separate mechanisms (surface capacitive behavior and diffusion-controlled Mg$^{2+}$ insertion processes): $i(V) = k_1v + k_2v^{1/2}$. Herein, $k_1v$ and $k_2v^{1/2}$ represent the capacitive- and diffusion-controlled current contributions, respectively. Figure 2e plots a capacitive current contribution at a CV scan of 10 mV s$^{-1}$, exhibiting a large percentage of capacitance (72.6%). In addition, Fig. 2f presents the bar chart of calculated capacitive contributions at various scan rates. Obviously, the ratio of capacitive contribution increases (from 60% to 98%) as the scan rate is raised from 1 to 100 mV s$^{-1}$, further verifying the capacitive-dominated reaction mechanism of the VN electrode.

**Probing the electrochemical reaction mechanism of VN.** To explore the reaction mechanism of the Mg$^{2+}$ ions on the VN electrode, operando XRD analysis was performed by using VN as the target electrode and MnO$_2$ as the counter electrode. All the XRD patterns were collected from the second cycle upon charge/discharge to avoid the irreversible reactions that occurred during the first cycle. Note that there are no new phases detected during the entire charge/discharge process. Figure 3a documents the obvious peak shift for the VN (200) plane (2θ: 44–45°). Upon charging (0–2.2 V), the diffraction peak gradually shifts toward high angles (2θ: 44.5–44.7°), indicating the contraction of the interplanar distance along the (200) direction. In response, the

diffraction peak returns to the low angle position upon discharging, indicating an expansion of the lattice spacing. The interplanar distance is calculated according the Bragg equation (λ = 0.15406 nm), to reversibly increase by 0.2 Å (2.1 → 1.9 → 2.1 Å) during the charge/discharge process (as depicted in Fig. 3b). Such a periodic variation in the lattice spacing confirms the reversible pseudo-capacitive Mg ion intercalation/de-intercalation of the VN (200) plane without causing a phase transition [11]. At the same time, the contraction of the interplanar distance during the Mg ion pseudo-capacitive intercalation can be explained by an increase of the electrostatic attraction between the guest Mg ions and the host VN lattice [12–14].

The pseudo-capacitive Mg ion intercalation mechanism of the VN electrode was verified by ex situ Mg 1s XPS measurements. As shown in Fig. 3c, the seven different sampling points are taken from a CV scan with a scan rate of 5 mV s$^{-1}$. The Mg 1s peak is centered at 1303.3 eV, which can be assigned to Mg$^{2+}$. As for the charging process (going from point 1 to 5), a gradual increase of the peak intensity can be clearly observed, reflecting the intercalation of Mg$^{2+}$ ions. During the discharging process (from −0.9 to 0 V), a decline in signal intensity signifies Mg$^{2+}$ de-intercalation. Furthermore, ex situ V 2p XPS spectra reveal the valence state change of vanadium upon charge/discharge (Supplementary Fig. 3). The peaks at 515.4 and 513.2 eV represent the contribution from V$^{3+}$ and V$^{2+}$, respectively [15], without the presence of any new signals. In this regard, one can conclude that V$^{3+}$ is the stable oxidation valence state in this VN electrode, whereas V$^{2+}$ is the relatively stable reduction valence state. It is worth pointing out that charge compensation will occur when the amount of positive charge in active materials increases [16]; in other words, during the insertion of Mg$^{2+}$ ions in the VN lattice. Figure 3d summarizes the proportion of vanadium valence states from the ex situ V 2p spectra. At the initial state, the concentration of V$^{3+}$ and V$^{2+}$ is 64% and 36%,

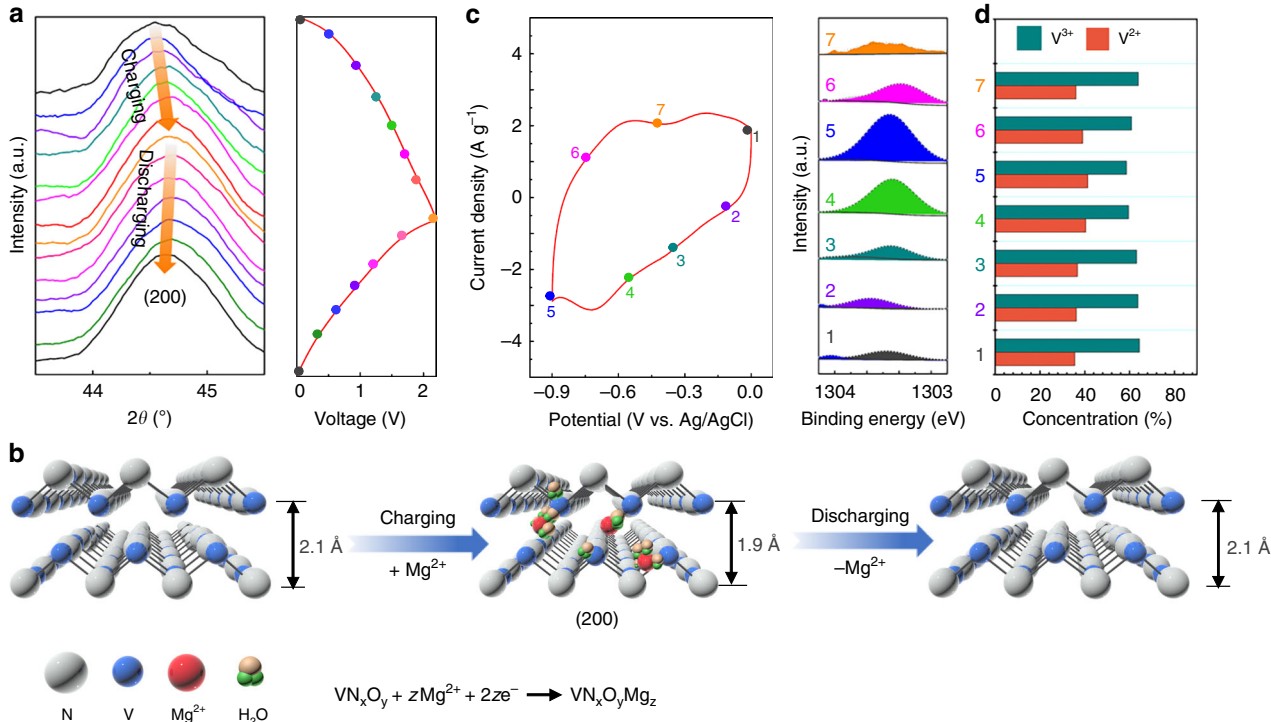

**Fig. 3** Electrochemical reaction mechanism of VN in a neutral $MgSO_4$ electrolyte. **a** Operando XRD patterns of VN during the charge/discharge process (current density: $0.1\ A\ g^{-1}$). **b** Schematic diagram showing the contraction and expansion of the (200) crystal plane during the charge/discharge process. **c** Ex situ XPS Mg 1s spectra at different potentials during the CV scan at a scan rate of $5\ mV\ s^{-1}$. **d** Change of vanadium valence state according to XPS V 2p spectra

respectively (point 1). In turn, the ratio of $V^{3+}$ decreases to 58% at the fully charged state (point 5). Upon full discharge, the proportion of $V^{3+}/V^{2+}$ returns back to 63%/37% (point 7), which is in accordance with the initial state. Considering the surface-sensitive nature of the XPS technique, these results reveal that the highly reversible surface redox reactions between $V^{3+}/V^{2+}$ also contribute to the charge storage mechanism during the electrochemical process.

**Evaluating performances of Mg ion quasi-solid-state ASCs.** Quasi-solid-state ASCs were assembled in typical coin cell configurations, where the $MnO_2$@carbon ($MnO_2$@C) composite was employed as the positive electrode for the insertion/extraction of $Mg^{2+}$ ions. The synthesized composites were subject to a wide suite of characterization tools (Supplementary Fig. 4). The three-dimensional $MnO_2$ nanoarchitecture boosts the Mg ion diffusion, while the conductive carbon matrix improves the electrical conductivity of the hybrid electrode. The electrochemical performances of the $MnO_2$@C composites were systematically measured in a three-electrode system (Supplementary Fig. 5). The electrode delivers a specific capacitance of $240\ F\ g^{-1}$ at a scan rate of $10\ mV\ s^{-1}$ with a broad potential window (from $-0.3$ to $1.2\ V$ vs. Ag/AgCl), which is similar to other cation electrolyte systems ($Li^+$, $Na^+$, and $K^+$). The $MnO_2$ electrode is likely to manifest a solid-state diffusion dominated charge storage mechanism, which is different from the capacitive charge storage behavior of the VN electrode (Supplementary Fig. 6)[17].

To assemble a quasi-solid-state ASC, the starch/PAM/$MgSO_4$ gel was introduced as the electrolyte[18]. In comparison with the commonly used polyvinyl alcohol, such a PAM-based gel matrix exhibits excellent compatibility with Mg cations. As expected, starch molecules and acrylamide molecules physically entangle and interpenetrate with each other to form an ion conductive network with favorable water absorption ability and mechanical robustness, as demonstrated in Fig. 4a. The as-prepared gel electrolyte can return to its initial state after various deformations, such as flatting, bending and/or stretching. Figure 4b presents CV curves for both a VN negative electrode and an $MnO_2$ positive electrode at a scan rate of $10\ mV\ s^{-1}$, indicating that the constructed asymmetric device can achieve a voltage window of 2.1 V or even larger. CV scans at different voltage ranges from 1.4 to 2.6 V were then carried out. Note that there is no obvious polarization even when the applied voltage is extended to 2.2 V (Fig. 4c, red line). Such a broad voltage range outperforms most ASCs reported recently[19,20]. Figure 4d shows the galvanostatic charge/discharge (GCD) curves of our quasi-solid-state ASCs at different current densities from 2 to $12\ mA\ cm^{-2}$. At $2\ mA\ cm^{-2}$, the discharge time reaches 637 s, corresponding to an areal capacitance of $576\ mF\ cm^{-2}$, which is 79.2% of the identical device tested in a liquid electrolyte (Fig. 4e; Supplementary Fig. 7). Interestingly, once the current densities exceed $4\ mA\ cm^{-2}$, the areal capacitance of the gel electrolyte is even higher than that of the liquid electrolyte-based system. The superior rate performance presented in gel electrolyte to that in the liquid electrolyte deserves further investigation. As such, comparative EIS profiles between the gel electrolyte and the liquid electrolyte systems before and after electrochemical tests have been collected (Fig. 4f; Supplementary Fig. 8). All these Nyquist plots have been fit according to a Randles equivalent circuit (Supplementary Table 2). It is evident that the $R_s$ of the gel electrolyte system only exhibits a negligible increase $(0.037\ \Omega)$ after electrochemical measurements, which is smaller than the $0.243\ \Omega$ of the liquid electrolyte system. The much steeper line at the low-frequency range in the Nyquist plots for the gel system (especially after the electrochemical tests) also indicates enhanced ion diffusion. Such a better electrolyte ion diffusion capability is the key to achieving

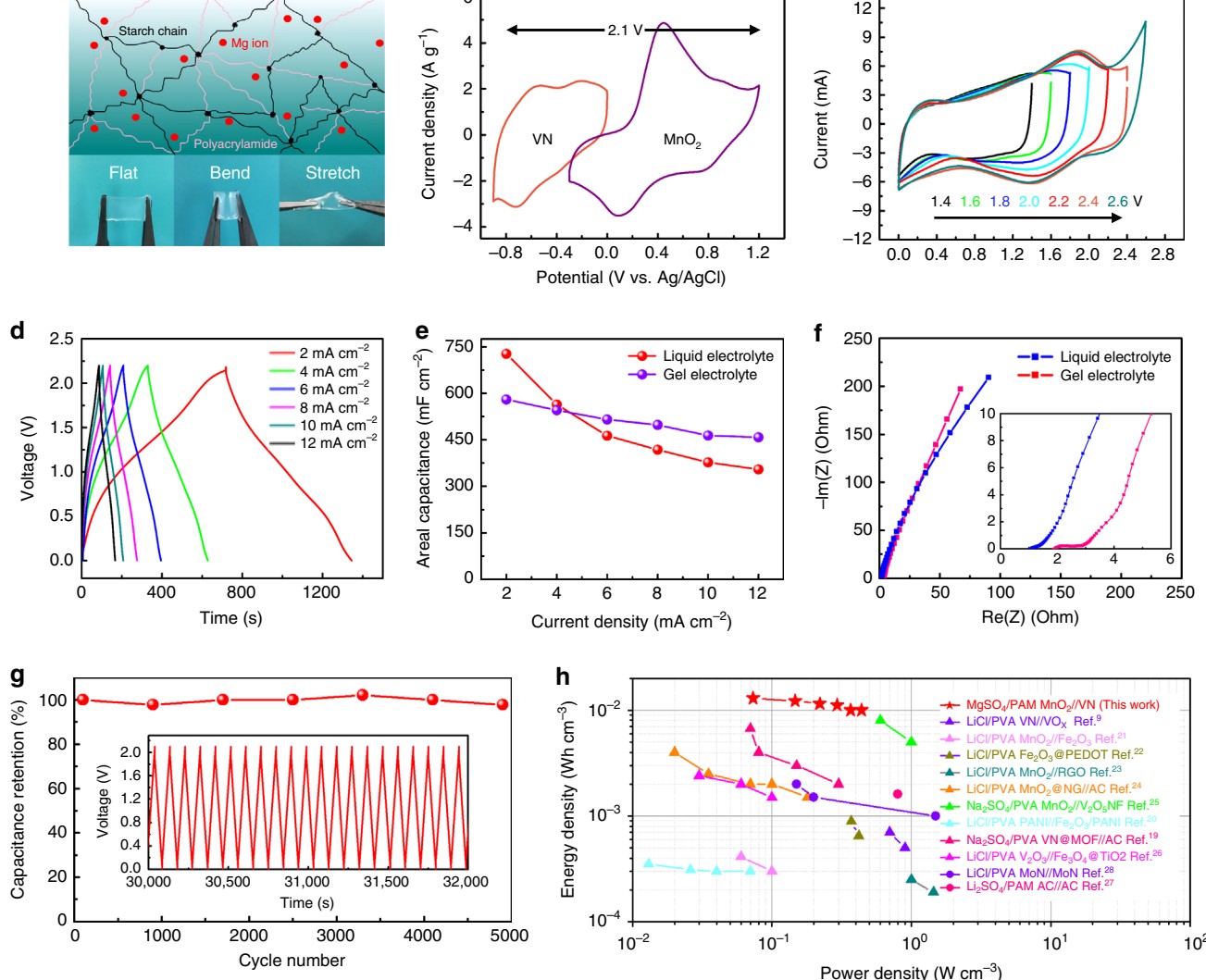

**Fig. 4** Electrochemical performance of Mg ion quasi-solid-state ASCs. **a** Structural diagram of a PAM gel electrolyte and digital photos showing its mechanical robustness upon deformation. **b** CV curve obtained for the VN and MnO$_2$ electrodes, respectively, at different potential ranges at a scan rate of 10 mV s$^{-1}$. **c** CV curves of a quasi-solid-state ASC with an increasing voltage window from 1.4 to 2.6 V at 10 mV s$^{-1}$. **d** Galvanostatic charge/discharge curves of ASCs under different current densities. **e** Comparison of areal specific capacitance of a liquid electrolyte and a gel electrolyte under different current densities. **f** Nyquist plot of a liquid electrolyte/separator and a gel electrolyte. **g** Long-term cycling stability of a quasi-solid-state ASC at a charge/discharge current density of 16 mA cm$^{-2}$. **h** Ragone plot of our quasi-solid-state ASC devices in comparison with other recently reported quasi-solid-state ASCs in neutral electrolytes

higher rate performance. Meanwhile, exhaustive morphological and elemental characterization provides evidence that the unique porous structure, high water content ratio and tight electrolyte/electrode interface contribute to the enhanced ion diffusion features of the gel electrolyte-based system (Supplementary Figs. 9–12). Notably, 95% capacitance retention can be achieved at a high current density of 16 mA cm$^{-2}$ after 5000 cycles, demonstrating an outstanding cycling stability of the assembled quasi-solid-state ASC (Fig. 4g). Furthermore, there is no distortion of the charge/discharge curves even for the last several cycles, as shown in the inset to Fig. 4g.

Figure 4h presents a Ragone plot of the volumetric energy density and the power density of our quasi-solid-state ASC (VN//MnO$_2$) in comparison with other recently reported quasi-solid-state supercapacitors tested in neutral gel electrolytes. Our device manifests a volumetric energy density of 13.1 mWh cm$^{-3}$ at a power density of 72 mW cm$^{-3}$. It also preserves 79% of its energy density when the power density is increased to 440 mW cm$^{-3}$.

These values are markedly superior to that of state-of-the-art systems[9,19–28], including VO$_x$//VN (LiCl/PVA)[9], MnO$_2$@NG//AC (LiCl/PVA)[24], VN@MOF//AC (Na$_2$SO$_4$/PVA)[19], and V$_2$O$_3$@C//Fe$_3$O$_4$@TiO$_2$ (LiCl/PVA)[26]. The high energy density and excellent rate performance of our device can be mainly attributed to the significantly extended stable voltage range of the Mg ion system and the favorable pseudo-capacitive Mg ion charge storage behavior of the VN negative electrode.

**Constructing flexible micro-asymmetric supercapacitors.** In pursuit of portable, miniaturized and wearable energy storage devices, we next fabricated flexible micro-asymmetric supercapacitors (MASCs) consisting of VN and MnO$_2$ interdigitated electrodes via screen printing[29–32] (Supplementary Fig. 13). Figure 5a depicts the general procedure for the direct printing of MASCs, including the pre-patterning of Au films as the current collectors[33], followed by deposition of interdigitated VN and

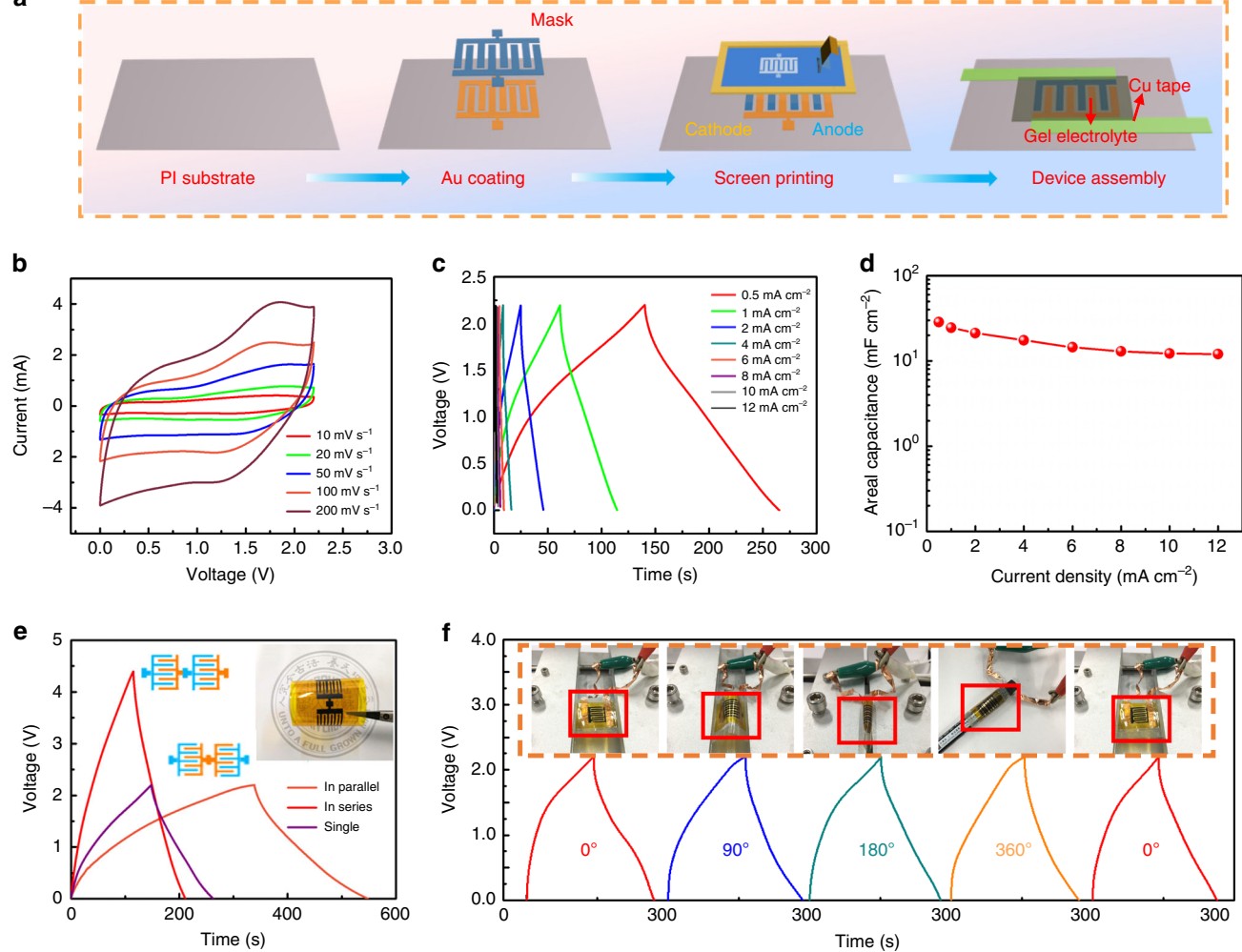

**Fig. 5** Electrochemical performance of a flexible and printable quasi-solid-state MASC. **a** Schematic illustration of the fabrication process of a MgSO$_4$/PAM VN//MnO$_2$ MASC. **b** CV curves of a printed MASC obtained at different scan rates. **c** Galvanostatic charge/discharge curves tested at different current densities. **d** Calculated areal specific capacitance at different current densities corresponding to **c**. **e** Galvanostatic charge/discharge curves obtained at 0.5 mA cm$^{-2}$ of single, series and parallel connections of two MASCs. **f** Galvanostatic charge/discharge curves tested at 0.5 mA cm$^{-2}$ under different bending angles

MnO$_2$ electrodes, and the application of the PAM-MgSO$_4$ gel electrolyte (with detailed dimensions of the screen-printing pattern shown in Supplementary Fig. 14). The electrochemical performances of the as-constructed VN//MnO$_2$ MASCs were characterized by CV and GCD measurements. Figure 5b shows the CV profiles of the MASC at different scan rates. Note that the open circuit voltage reaches a remarkable 2.2 V, exceeding most of the recently reported in-plane micro-supercapacitors and ASCs. A couple of redox peaks can still be detected, in good agreement with the foregoing results of the quasi-solid-state asymmetric SC in a coin cell configuration. The CV shape is well preserved from 10 to 200 mV s$^{-1}$, suggesting excellent rate capability of the device. Figure 5c displays the GCD curves at different current densities from 0.5 to 12 mA cm$^{-2}$, showing obvious triangular shapes. As such, our printed MASC delivers an areal capacitance of 28.5 mF cm$^{-2}$ at 0.5 mA cm$^{-2}$, while 42% of this capacitance remains upon testing at 12 mA cm$^{-2}$, as shown in Fig. 5d. The thus-derived MASCs display a slow self-discharge process (Supplementary Fig. 15) and a stable long cycle performance (a capacitance retention of 90% after 8000 cycles; see Supplementary Fig. 16). In addition, the Ragone plot drawn in Supplementary Fig. 17 demonstrates that our printed MASCs

harvest an energy density of 19.13 μWh cm$^{-2}$ at a power density of 0.55 mW cm$^{-2}$, outperforming those of recently reported in-plane micro-supercapacitors (see Supplementary Table 3 for detailed information).

To efficiently boost the operating voltage and capacitance output of a single device, MASCs could be easily connected in series or in parallel by optimizing the printing process. Figure 5e presents the GCD curves of single, series and parallel MASCs. In this regard, a series connection enables the extension of the output voltage to 4.4 V (double the 2.2 V for a single device). Meanwhile, the capacitance can be readily doubled by adopting a parallel connection, demonstrating advanced modulation for device integration[33]. To further evaluate the mechanical robustness of our MASC targeting practical wearable applications, we carried out GCD measurements under bending angles of 0°, 90°, 180°, and 360° (Fig. 5f, and inset). Notably, the shapes of the GCD curves with different bending angles only change slightly, with almost 100% of the initial capacitance retained on bending at 360°, thus demonstrating excellent mechanical flexibility. Such printed MASCs with their excellent mechanical properties are thus conducive to multi-field integration with solar cell systems.

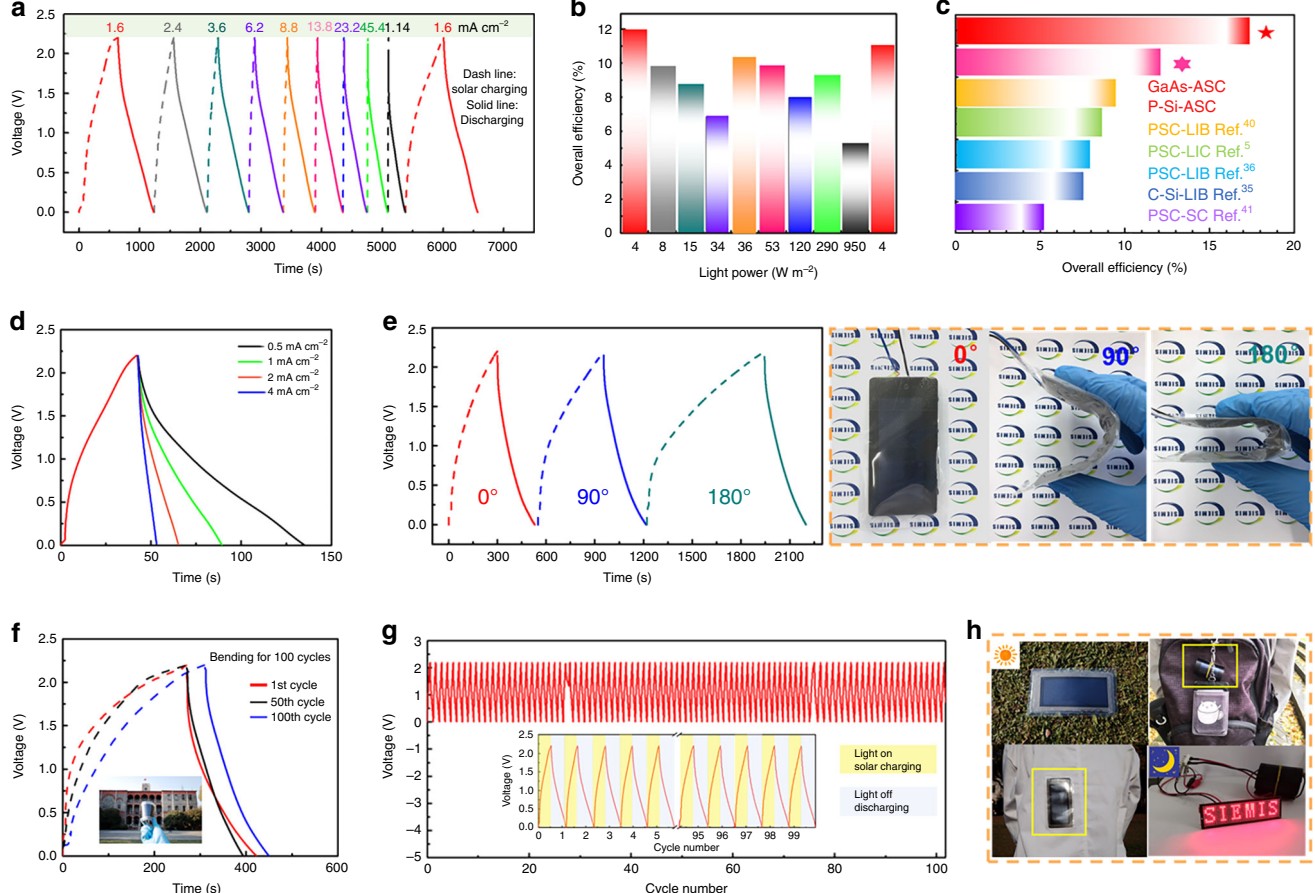

**Fig. 6** Electrochemical performance of as-constructed solar-charging integrated units. **a** Solar-charging/discharging curves at different light intensities and the same discharge current density of 1 mA cm$^{-2}$. **b** Calculated $\eta_{overall}$ at different light intensities corresponding to **a**. **c** Comparison of $\eta_{overall}$ with a recently reported solar-charging integrated energy system. **d** Voltage–time profiles of our flexible solar-charging integrated units. **e** Solar-charging/discharging curves at different bending angles (left panel), with digital photos (right panel) showing bending conditions. **f** Durability test of a flexible solar-charging integrated unit for 100 cycles. **g** Long cycling stability of a flexible solar-charging integrated unit. **h** Demonstration of wearable scenarios using our solar-charging integrated units

**Integrating flexible solar-charging self-powered units**. The rapid development of wearable smart electronics has led to a huge demand for flexible, safe, and long-life energy sources[34]. The printed MASCs as demonstrated above could serve as viable energy storage devices, but might still not meet the self-powered requirements with no external charging supply, especially for wearable applications. In this respect, integration with an energy-harvesting component, such as a solar cell, is a feasible solution to both capture and energy storage within a single device. We started by evaluating our flexible VN//MnO$_2$ ASC within a solar-charging integration system. To achieve this, a low-cost poly-crystalline Si solar cell was employed as the PV module (Supplementary Fig. 18a, b) and an as-assembled quasi-solid-state ASC was used as the energy storage module. Figure 6a displays the solar-charging/discharging profiles of the integrated device under varied light intensities with the same discharge current density (1 mA cm$^{-2}$). The output current density of the solar cell corresponding to the light intensity is shown in Supplementary Fig. 18c. When the light intensity was increased from 4 to 950 W cm$^{-2}$, the output current density responsively increased from 1.6 to 114 mA cm$^{-2}$. Such an abrupt increase in light intensity (current) could significantly shorten the duration of solar-charging (e.g. to 3 s). In addition, the discharge time is correspondingly reduced, which can be attributed to an increased IR drop at the large current densities. It is worth-noting that the

discharge behavior can still return to the initial state upon dialing the light intensity back to 4 W cm$^{-2}$, indicative of excellent rate capability. In other words, our ASC demonstrates the ability to withstand large currents (e.g., 114 mA cm$^{-2}$), outperforming emerging energy storage counterparts (such as lithium ion batteries, LIBs[35,36], and capacitors, LICs[5]). The overall efficiency for energy conversion and storage ($\eta_{overall}$) has been calculated at different light intensities, as summarized in Fig. 6b. The highest $\eta_{overall}$ is achieved at a light intensity of 4 W cm$^{-2}$ (corresponding to typical indoor-ambient lighting conditions), reaching a remarkable 11.95%. Such a high overall efficiency and excellent indoor light performance can be attributed to the high current tolerance and high-energy storage efficiency ($\eta_{storage}$) of our ASC, where the $\eta_{storage}$ is calculated to be ~80%. To further verify the superiority of our solar-charging system, a GaAs-ASC integrated device was fabricated by combining GaAs solar cells with assembled ASCs, demonstrating a record high $\eta_{overall}$ of 17.57% under 1000 W m$^{-2}$ (Supplementary Fig. 19). Such a remarkable overall efficiency can be attributed to the high conversion efficiency of the GaAs solar cell (25.88%) and the high-energy storage efficiency of the ASC (67.90%). Figure 6c draws comparisons of $\eta_{overall}$ between our integrated unit and state-of-the-art solar-charging systems. It is obvious that the $\eta_{overall}$ of our Si solar cell/ASC and GaAs solar cell/ASC devices outperform the other integrated systems, such as a Si solar cell integrated LIB (c-Si-LIB,

7.61%) and a perovskite solar cell combined with a LIC (PSC-LIC, 8.41%). Detailed comparisons of the key parameters (overall efficiency, output voltage and endurance current) between the current work and other reported studies in the field of solar-charging self-powered systems is provided in Supplementary Table 4.

We then considered the practical uses of our system by assembling a flexible solar-charging integrated unit according to the configuration depicted in Fig. 1a. Notably, a flexible amorphous Si solar cell functioned as the PV module (Supplementary Fig. 20a, b). The VN//MnO$_2$ MASC was printed directly on the bottom of the solar cell, which is buffered by a polyimide (PI) layer. Figure 6d illustrates the solar-charging/discharging curves of our flexible integrated device. Under the same solar-charging condition of 4 W cm$^{-2}$, it exhibits a favorable rate performance. The $\eta_{overall}$ of such a flexible system reaches 5.2% (Supplementary Fig. 20c). Despite being not as high as our rigid integrated unit, this is still superior to most reported flexible photo-charging self-powered systems[4,37–41] (see Supplementary Table 5). To envisage realistic wearable applications, we further evaluated the mechanical stability of the integrated solar-charging self-powered units. Figure 6e shows the solar-charging (4 W cm$^{-2}$)/ discharging (1 mA cm$^{-2}$) profiles under bending angles of 0°, 90°, and 180° (as displayed by the digital photos). The observation of increased solar-charging duration as a function of bending can be mainly attributed to the decrease in the effective area of illumination of a bent solar cell. Under this circumstance, the discharge energy, indicated by the discharging time, remains constant, demonstrating good flexibility of our self-powered unit. A mechanical durability test was also carried out by repeatedly bending the integrated device at 90° for 100 cycles (Fig. 6f,inset). The capacitance retention reached 94% and 80% after 10 and 100 bending cycles, respectively. In further tests, various flexible conditions were implemented to verify the mechanical robustness of our devices (Supplementary Fig. 21). Before and after folding, rolling, and waving deformations, the integrated unit exhibited a capacitance retention of 97.50%, 85.22%, and 93.04%, respectively. All these provide evidence for the good flexibility of our self-powered unit for practical applications.

Figure 6g further presents the photo-charging/discharging cycling performance tested at a light intensity of 4 W cm$^{-2}$ and a discharge current density of 1 mA cm$^{-2}$, with the inset showing the first 10 cycles and last 10 cycles of photo-charging/ discharging curves. Remarkably, our integrated unit maintained outstanding cycling stability with a surprisingly high capacitance retention of 98.7% after 100 cycles, thus demonstrating excellent robustness and mechanical stability. It is anticipated that our integrated solar-charging self-powered units with good flexibility, long cycle life, and high safety will be viable for many wearable scenarios. Figure 6h shows a proof-of-concept demonstration. The integrated unit can be fully charged when exposed to either natural sunlight or outdoor lighting (i.e., during the daytime). The flexible and even rollable feature of such a power unit enables the direct wearing on clothes or application to bags as an efficient self-powered energy source. The energy stored in the solar-charged MASC can be ultimately utilized to power a light-emitting-diode panel, especially for indoor use during night time.

## Discussion

In summary, a highly flexible, safe, and durable solar-charging self-powered integrated unit has been developed. By synergistically combining an in-plane asymmetric printing technology with a multivalent ion electrochemical energy storage technology, the thus-derived integrated units show outstanding photo-charging

cyclic stability (up to 98.7% capacitance retention after 100 cycles), with tolerance for a wide variation in current (from 1.6 to 114 mA cm$^{-2}$), and favorable energy management for wearable devices. Moreover, the pseudo-capacitive intercalation/de-intercalation reaction mechanism of Mg$^{2+}$ ions in VN electrodes has been elucidated using operando XRD and ex situ XPS, explaining both high energy density (13.1 mWh cm$^{-3}$) and high power density (440 mW cm$^{-3}$) of the VN//MnO$_2$ ASCs. Such ASCs based on multivalent neutral ion electrolytes offer new insights into the design of high-energy, long-life, and safe energy sources toward the ultimate goal of creating light weight and wearable smart electronics.

## Methods

**Preparation of VN and MnO$_2$@C electrode materials.** Ammonium metavanadate was dissolved in 135 mL of deionized water and 15 mL of ethanol by stirring, followed by the addition of hydrochloric acid (1 mol L$^{-1}$) to adjust the pH value of the mixture to 2.0. The mixture was then transferred into a Teflon-lined autoclave and kept at 180 °C for 24 h. After cooling down to room temperature, a nanobelt-like VO$_2$ product was obtained upon washing and drying. Finally, VN nanowires were produced by annealing the above-synthesized VO$_2$ in pure NH$_3$ at 600 °C for 3 h. As for the MnO$_2$@C synthesis, 2 mL of an aqueous GO suspension (5 mg mL$^{-1}$) was added into 58 mL of deionized water, followed by the addition of KMnO$_4$ (316 mg) and urea (1 g). After stirring for 20 min, the above solution was transferred into a Teflon-lined autoclave and kept at 120 °C for 12 h. Nanoflower-like MnO$_2$@C was readily obtained after washing and drying.

**Preparation of MgSO$_4$-PAM gel electrolyte.** Three grams of starch was dissolved into 30 mL of deionized water by vigorously stirring at 100 °C for 1 h. After cooling, 5 g of acrylamide monomer, 25 mg of NH$_4$S$_2$O$_8$, and 3 mg of $N,N'$-methylenebisacrylamide were added. The solution was stirred at room temperature for 2 h, followed by the elimination of surface bubbles under vacuum. The solution was then poured into a mold of a specific thickness and heated at 80 °C for 1 h. The as-prepared PAM gel was then immersed into 1.0 M MgSO$_4$ electrolyte for more than 24 h to reach the ion balance point.

**Assembly of quasi-solid-state ASCs.** Mg ion quasi-solid-state ASCs were assembled using a typical coin cell structure. The negative electrode was prepared by casting onto Ti mesh containing the active material VN, conductive carbon (Super P), and binder (PVDF) in a mass ratio of 8:1:1. The mass loading was adjusted to 5–6 mg cm$^{-2}$. The MnO$_2$@C positive electrode was fabricated using a similar method, with the mass ratio at 7.5:2:0.5. The MgSO$_4$-PAM gel was employed as the electrolyte. The optimized mass ratio of the positive and negative electrode was 1:2 (Supplementary Fig. 22). The thickness of the negative electrode, the positive electrode, and the gel electrolyte were 50, 50, and 200 μm, respectively. Moreover, a liquid electrolyte-based ASC was assembled as a control by using glass fiber as the separator with a liquid electrolyte.

The capacitance is calculated using the following equations:
According to the CV curves

$$C = \frac{1}{\nu * \Delta V} \int_{V1}^{V2} I(V) \mathrm{d}V. \tag{2}$$

In the three-electrode system, $C$ is the capacitance (F), $\nu$ is scan rate (V s$^{-1}$), $I$ is the current response (A), $V1(V2)$ is the starting and ending voltage (V), and $\Delta V$ is the voltage window (V). For the single electrode, the specific capacitance is calculated according the following equation:

$$C_m = \frac{C}{2m}, \tag{3}$$

where $C_m$ is the mass specific capacitance (F g$^{-1}$), $C$ is the capacitance (F), and $m$ is the mass of active material (g; without the consideration of binders and conductive agents).

In the two-electrode system, the electrochemical performance is calculated according to the GCD curves

$$C = \frac{It}{\Delta V}, \tag{4}$$

where $C$ is the capacitance (F), $I$ is the discharging current (A), $t$ is the discharging time (s), and $\Delta V$ is the voltage window (V). For the device, the specific capacitance is calculated according the following equation:

$$C_m(s) = \frac{C}{m(s)}, \tag{5}$$

where $C_m(s)$ is the mass or area specific capacitance (F g$^{-1}$ or F cm$^{-2}$), $m$ is the total mass of positive and negative active material (g), $s$ is the area of positive or

negative electrode ($cm^{-2}$, the area of positive and negative electrode is the same), and $C$ is the capacitance (F).

The energy density ($E$) and power density ($P$) can be derived according to the following equations:

$$E_m = \frac{1}{2 \times 3.6} C_m \times V^2, \qquad (6)$$

where $E_m$ is the mass energy density (Wh $kg^{-1}$), $C_m$ is the mass specific capacitance (F $g^{-1}$), and $V$ is the voltage (V).

$$E_v = \frac{1}{2 \times 3600} C_v * V^2, \qquad (7)$$

where $E_v$ is volume energy density (Wh $cm^{-3}$), $C_v$ is the volume specific capacitance (F $cm^{-3}$, the volume is the product of electrode area and thickness, which includes the thickness of the positive electrode, the gel electrolyte, and the negative electrode), and $V$ is the voltage (V).

As for the power density, the calculated equation is

$$P = \frac{E}{t}, \qquad (8)$$

where $P$ is the power density (W $kg^{-1}$ or W $cm^{-3}$), $E$ is the energy density (Wh $kg^{-1}$ or Wh $cm^{-3}$), and $t$ is the discharging time (h).

**Construction of flexible MASCs via printing.** Flexible MASCs were constructed via a sequential screen-printing method. First, the PI substrate was treated with an $O_2$ plasma. The as-treated substrate was subjected to a DC sputtering of gold for 15 min, where a printing mask was used to outline the gold deposition. Next, the VN electrode was printed with a slurry containing 80% VN, 10% conductive carbon black and 10% LA132. After drying, a second printing was applied to create the $MnO_2$ electrode (75% $MnO_2$@C, 15% Super P, and 10% LA132). A gel electrolyte was then coated over the patterned electrodes, with copper tape employed as the current collectors. Finally, the entire device was wrapped in a commercial thermoplastic PET film (see Supplementary Note 2 for the printing details). The encapsulation process is described as follows: First, the Cu conductive tape was pasted over the lead, the gel electrolyte was then coated on the printed pattern. Second, the micro-supercapacitor was placed between two plastic films and sealed by thermoplastic sealing. Finally, the tip of the lead was fixed using an AB glue (Supplementary Fig. 23).

**Assembly of flexible solar-charging integrated units.** To assemble the flexible self-powered units, flexible commercial amorphous Si solar cells were used as the energy-harvesting module. To avoid the non-smooth surface of the flexible solar cell, acrylic double-sided tape was first pasted onto the backside of the solar cell. Then, the cleaned PI substrate was attached to the other side of the acrylic tape, which was used for the printed substrate of the MASCs. After that, the MASC was fabricated by screen-printing. The gel electrolyte was then coated on the micro-pattern and the MASC was sealed using plastic film. Finally, the self-powered unit was packaged using the same plastic film (see Supplementary Fig. 24). To reach the matched open circuit voltage, two solar cells were connected in series to power a MASC module.

The overall efficiency of energy conversion and storage ($\eta_{overall}$) can be calculated according to the following equation:

$$\eta_{overall} = \frac{E}{P \times S \times t} \times 100\%, \qquad (9)$$

where $E$, $P$, $S$, and $t$ are the discharge energy after solar-charging (Wh), the light intensity (W $m^{-2}$), the effective area of solar cell ($m^2$), and the duration of solar-charging (h), respectively.

**Characterization.** The morphologies of samples were characterized by a Hitachi SU8010 scanning electron microscopy. STEM and HRTEM images were obtained on a Cs-corrected FEI Titan transmission electron microscopy operated at 300 kV. XPS spectra (including ex situ XPS analysis) were collected using an Escalab 250Xi Spectrophotometer using a monochromatic Al-Kα X-ray source. XRD measurements (including operando XRD) were performed at a Bruker D8 Advance Diffractometer using Cu-Kα radiation ($\lambda = 1.5406$ Å). All the electrochemical data were measured by a CHI760E workstation (CH Instruments, USA). The three-electrode tests (CV, LSV, GCD) were performed utilizing an Ag/AgCl reference electrode and a graphite rod counter electrode.

## Data availability

The data supporting the findings of this work are available within the article and its Supplementary Information files. All other relevant data supporting the findings of this study are available from the corresponding author on request.

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

## Acknowledgements

This work was financially supported by the National Natural Science Foundation of China (51702225), National Key Research and Development Program (2016YFA0200103), and Natural Science Foundation of Jiangsu Province (BK20170336). The authors also acknowledge support from the Suzhou Key Laboratory for Advanced Carbon Materials and Wearable Energy Technologies, Suzhou, China and the Dr. Myung Ki Hong Chair in Materials Innovation (to R.B.K.).

## Author Contributions

J.S., Y.S., R.K. and Z.L. conceived and designed the experiments. Z.T. and X.T. prepared the electrode materials and carried out the characterization, fabricated the MgSO$_4$/PAM gel electrolyte, and fabricated the integrated self-powered device and investigated its photo-charging/discharging properties. Z.T. probed the reaction mechanisms using operando XRD and ex situ XPS. Z.T. fabricated the MASCs via printing, with L.Y.'s assistance. G.S., Y.S. and V.T. performed HRTEM and STEM mapping measurements. Z.T., Y.S., J.S., R.B.K. and Z.L. were mainly responsible for preparing the manuscript with input from all other authors. All authors discussed the results and commented on the manuscript. All authors have given approval to the final version of the manuscript.

## Competing Interests

The authors declare no competing interests.
