## [Peer Review File · Nature Communications]

Reviewers' comments:

Reviewer #1 (Remarks to the Author):

The authors have reported a flexible self-powered unit by combining the printed micro-supercapacitor and thin-film solar cell. The design of the Mg-ion based aqueous supercapacitor system brings about a capacitive-dominated energy storage mechanism and extended energy density, which lead to long cycling capability and high energy conversion efficiency of the overall integrated solar-charging unit. In addition, the Mg-ion based pseudocapacitive charge storage mechanism in aqueous electrolyte was studied in detail by operando XRD and ex situ XPS. This work is interesting and will provide insights for the field of flexible energy storage, printed electronics, etc. So I recommend its publication after the authors have carefully addressed the following issues:

1. High energy density of supercapacitors is significant for wearable electronics. However, this work only reported the energy density of Mg-ion quasi-solid-state ASCs. How about that of the printed micro-asymmetric supercapacitors?
2. The authors used both "capacitance" and "capacity" throughout the manuscript which could lead to misunderstanding for readers. The authors should clarify this point.
3. Why using nanowire-shaped VN as the electrode? The authors used different characterizations on VN materials in manuscript. Thus the authors should give more explanation of this point in the Introduction part.
4. Why the authors put Li^+ in the Faradaic reactions of VN in line 150? VN electrodes only show redox reaction in Mg^{2+} electrolyte. In Figure 2A, there are two oxidation peaks while only one reduction peak, can the authors explain this?
5. Equivalent circuit fitting should be provided in Figure S2.
6. The authors should explain why the capacitive contribution part exceeds the total CV curve in Figure 2E.
7. Although the authors conducted the operando XRD to prove the VN lattice shrinking during the Mg^{2+} intercalation, this phenomenon still needs more explanation.
8. Figure captions in Figure 3B-D are not correct.
9. The authors used VN and MnO_2 as negative and positive electrode, respectively. How did the authors balance the charges of these two electrodes in the quasi-solid-state ASCs and printed MASC?
10. It's unusual that the gel electrolyte-based system exhibited better rate performance and electrolyte ion diffusion behaviour than the liquid electrolyte-based system, as shown in Fig. 4 E-F. Why the intrinsic equivalent series resistance is larger in the PAM gel system but the Mg^{2+} diffusion is better?
11. The author should provide more details of electrochemical performance calculations in the experimental section.
12. The printed MASC which was used as the energy storage unit was integrated with PV in a solar-charging self-powered system, thus the self-discharge of MASC is an important parameter for evaluation of this self-powered system. What is the self-discharge performance of printed MASC?
13. The bending test of flexible integrated solar-charging self-powered units seems not enough for practical application. The capacitance retention reached 74 % after 20 cycles. Can authors improve the capacitance retention after bending test?
14. The line width of the interdigitated electrodes and the pitch between two fingers have great influence on the performances of the micro-asymmetric supercapacitors. Please give these values.
15. A plastic film was used to wrap the device but what is the plastic film. As the long-term stability (specific capacitance changing with time) is essential for the practical application of the micro-asymmetric supercapacitors, thus supplement the related tests.
16. The assembly of flexible solar-charging integrated units, especially the interconnect between the MASC and the Si solar cells is ambiguous. Please give detailed descriptions of the structure of this integrated system in the text and supplement the detailed fabrication process in the experimental section.
17. To catch broad interest, the background of printed supercapacitors and flexible supercapacitors

should be discussed extensively. Some related papers are suggested for references, such as: Chemical Society Reviews, 2019, 48, 3229-3264; Advanced Materials Technologies, 2019, 1900196; Advanced Materials, 2016, 28, 5242-5248; etc.

Reviewer #2 (Remarks to the Author):

This manuscript (NCOMMS-19-19708) describes solar charging self-powered units using printable Mg-ion quasi-solid-state asymmetric supercapacitors. The idea of "self-powered & integrated devices" is a current hotspot and has good application prospects. The written English is commendably good, but there are still some deficiencies. In general, the manuscript could be accepted after major revision.

1. MnO₂ and VN possess different charge storage abilities and the mass ratio of the MnO₂ cathode to the VN anode will affect the specific capacitance and energy density of assembled MnO₂//MgSO₄-polyacrylamide//VN ASCs devices. The mass ratio is the most critical factor for asymmetric devices. Please provide more information about the influence of the mass ratio of the anode and the cathode on the electrochemical performance of the device. In addition, how to control the screen-printing process effectively?

2. Encapsulating material is important for flexible electronics, what is the encapsulating film for the integrated unit? How to ensure that the encapsulating material is impervious to water and oxygen during long-term testing? Detailed description of the packaging process is necessary.

3. The authors used "crosslink with each other" to describe starch molecules and acrylamide molecules (page 10 line 252), this is a misunderstanding of the concept of crosslink. In fact, crosslinking only occurs among acrylamide molecules chains; for starch molecules and acrylamide molecules chains, physical entanglement and interpenetration between molecular chains are the correct mechanism.

4. The explanation of the better rate performance for the gel electrolyte compared with the liquid electrolyte-based system (Fig. 4E) is somewhat far-fetched, more convincing explanation or experiments are necessary.

5. Why use "mF cm⁻²" instead of keeping up with "F g⁻¹" of a single electrode when evaluating electrochemical performance of coin cell quasi-solid-state ASCs? Comparing the electrochemical performances of the assembled devices with a single electrode can embody the importance of mass ratio.

6. The title is about "flexible solar-charging integrated units", but the manuscript falls behind the title. A large part of the text is about the characterization of anode materials VN, yet the description about "flexible self-powered" is not substantial, only simple bending test was mentioned, other flexible characterizations such as twisting, folding, knotting, etc. were not implemented. Some characterizations of the anode material could be put in the supplementary information, since we are more interested in "flexible self-powered" and its application in realistic situations.

7. Other errors, such as extra space (page 15 line 409), inappropriate molecular formula (Supplementary Figure 1a), also exist. Please double check the manuscript.

Reviewer #3 (Remarks to the Author):

In this report, Tian et al. reported the Mg-ion induced microsupercapacitors with self-charging solar-energy storage units for portable electronic applications. The on-chip microsupercapacitors were fabricated using vanadium nitride and manganese oxide-based materials, which demonstrated an energy density of 279 13.1 mWh cm⁻³ at a power density of 72 mW cm⁻³ with good cycling stability. Following, a flexible solar cell was assembled with microsupercapacitor as self-charging units, which demonstrated a maximum energy conversion/storage efficiency of 11.95%. It would be interesting finding, if the integrated device was assembled in the same housing by developing designs where

electrodes are shared between the energy storage unit and the solar cell. In the current work, the authors have designed an externally connected microsupercapacitor with solar cell, which creates external ohmic voltage loss and leakage current. Considering the significance level of the prepared materials and the reported approach in the fabrication of microsupercapacitors and self-charging units are not new, as this kind of approaches have already been reported earlier for various portable applications, such as Wang et al, *npj 2D Materials and Applications*, 2, 7, 2018, Etienne et al., *Electrochem. Commun.* 28, 104, 2013, Maher et al, *Proc. Natl. Acad. Sci.*, 2015, 112, 4233, Gao et al, *Nat. Comm.*, 7, 11586, 2016, Yun et al. *Nano Energy*, 49 (2018), 644, Ahmad et al, *Nano Lett.* 2018, 18, 1856, Kim et al, *J. Mater. Chem. A*, 2017, 5, 1906). Therefore, I do not recommend this manuscript to be suitable for publication in a well-reputed journal like Nature Communications.

Revision Summary

- 1) According to Referee 1's comment with respect to the energy density of printed MASCs, we have included the calculated energy density value and drawn a comparison with other reported in-plane supercapacitor devices (**Supplementary Fig. 17; Supplementary Table 3**).
- 2) Based on the comment from Referee 1, we have added an explanation pertaining to the advantages of the nanowire-shaped VN electrode. We have provided the equivalent circuit fitting as suggested in the revision (**Supplementary Fig. 2; Supplementary Table 1 & Note 1**). We have also corrected the figure captions for **Fig. 3B-D**.
- 3) As requested by Referee 1 and Referee 2, we have added a detailed discussion on the charge balance and loading mass ratio control of ASC devices (**Supplementary Fig. 22; Supplementary Note 2**), and further illustrated the packing and sealing process of integrated devices (**Supplementary Figs. 14, 23, 24**).
- 4) As requested by Referee 1 and Referee 2, we have carried out supplementary tests in order to discuss the enhanced rate performance of the device in the gel electrolyte system (**Supplementary Figs. 8-13**).
- 5) Based on the suggestion from Referee 1, we have additionally tested the self-discharge behavior (**Supplementary Fig. 15**) and long-term cycling stability (**Supplementary Fig. 16**) of our MASCs. We have also added the detailed calculation methods used to evaluate electrochemical performances.
- 6) As requested by Referee 1, we have incorporated the suggested literature as **Refs. [30][31][32]**.
- 7) Based on the comment from Referee 2, we have revised **Fig. 1** in the main text by putting some parts of the VN characterization results into the supplementary information. More importantly, we have carried out additional measurements to demonstrate the flexible capability of our "flexible self-powered unit" (**Fig. 6F; Supplementary Fig. 21**).
- 8) As pointed out by Referee 2, we have corrected the typos and inappropriate phrases in the revised manuscript (e.g., starch molecules and acrylamide molecules physically entangle and interpenetrate with each other...).
- 9) According to Referee 3's feedback, we have provided a detailed explanation on the novelty of the current work, and further incorporated the related discussions into the revision (**Fig. 6C; Supplementary Fig. 19; Supplementary Tables 4 & 5, as well as the discussion therein**).

Below we provide point-by-point responses to the referees' comments in detail and point out the places where we have revised the text in the manuscript.

Referee: 1

Referee 1 (Remarks to the Author):

The authors have reported a flexible self-powered unit by combining the printed micro-supercapacitor and thin-film solar cell. The design of the Mg-ion based aqueous supercapacitor system brings about a capacitive-dominated energy storage mechanism and extended energy density, which lead to long cycling capability and high energy conversion efficiency of the overall integrated solar-charging unit. In addition, the Mg-ion based pseudocapacitive charge storage mechanism in aqueous electrolyte was studied in detail by operando XRD and ex situ XPS. This work is interesting and will provide insights for the field of flexible energy storage, printed electronics, etc. So I recommend its publication after the authors have carefully addressed the following issues:

Response

We thank the referee for these positive and helpful remarks. Our response follows each comment.

1. High energy density of supercapacitors is significant for wearable electronics. However, this work only reported the energy density of Mg-ion quasi-solid-state ASCs. How about that of the printed micro-asymmetric supercapacitors?

Response

We appreciate the instructive comment. We fully agree with the referee that the energy density of printed micro-supercapacitors should be further discussed. Accordingly, we have calculated/summarized the energy density and other electrochemical performances of our $\text{MnO}_2@\text{C}/\text{VN}$ micro-asymmetric supercapacitor (MASC), and further drawn performance comparisons between our device and other reported in-plane micro-supercapacitors in the literature (Figure R1; Table R1). Our $\text{MnO}_2@\text{C}/\text{VN}$ MASC device exhibits a wide stable working voltage and favorable energy/power density performance. The stable operating voltage is comparable with those based on the organic electrolyte system (Ref. 1) and even higher than

those of the ionogel electrolyte system (Ref. 3). The energy density and power density of our current work outperforms all the listed published studies.

Figure R1. Ragone plot of the MnO₂@C//VN micro-supercapacitor in comparison with other recently reported micro-supercapacitors.

Table R1. Electrochemical Performance Comparison of Recently Reported In-plane Supercapacitor Devices

System	Electrolyte (gel)	Voltage (V)	Energy (μWh/cm ²)	Power (mW/cm ²)	Ref.
MnO ₂ @C//VN	MgSO ₄ /PAM	0-2.2	19.13	22.96	This work
3D Porous C	LiTFSI	0-2.5	4.9	7.92	1
NiCoP@NiOOH//ZIF	KOH/PVA	0-1.4	13.9	2	2
Cu(OH) ₂ @FeOOH/Cu	[EMIM][BF ₄]/SiO ₂	0-1.5	18.07	0.7	3
3DGN/SWNT/AgNW	LiCl/PVA	0-1.0	2.75	0.361	4
Ppy@MWCNT//MnO ₂ @Ppy	LiCl/PVA	0-1.6	12.16	17.65	5
MoS ₂ @rGO-CNT	H ₂ SO ₄ /PVA	0-1.0	2	1	6
3D graphene	H ₂ SO ₄ /PVA	0-1.0	0.38	14.4	7
Ti ₃ C ₂ T _x	H ₂ SO ₄ /PVA	0-0.5	0.32	0.158	8
MnO ₂ //PPy	CMC-Na ₂ SO ₄	0-1.5	8.05	7.32	9

Following this comment, we have revised the relevant contents in the revised manuscript (**Lines 317-319; Supplementary Fig. 17; Supplementary Table 3**).

“Ragone plot drawn in Supplementary Fig. 17 demonstrates that our printed MASCs harvest an energy density of $19.13 \mu\text{Wh cm}^{-2}$ at a power density of 0.55 mW cm^{-2} , outperforming those of recently reported in-plane micro-supercapacitors (see Supplementary Table 3 for detailed information).”

2. The authors used both “capacitance” and “capacity” throughout the manuscript which could lead to misunderstanding for readers. The authors should clarify this point.

Response

We thank the referee for this comment. The difference between capacity and capacitance is derived from the different definitions and related charge-storage mechanisms. For battery-type energy storage devices, there is a relatively flat charge/discharge plateau at a constant voltage stage. In this case, the ability to store charge is based on a Faradaic reaction and is called “capacity”, which is in units of mAh. In contrast, the capacitor-type energy storage device exhibits a sloping shape with a constant slope value during the galvanostatic charge/discharge process. Then the ability to store charge within a specific voltage window is called “capacitance”, which is in units of F (Farads). With regard to our present work, the charge storage mechanism is mainly dominated by capacitive behavior and the galvanostatic charge/discharge curves exhibit typical oblique lines. Hence, we modified all descriptions in this manuscript from “capacity” to “*capacitance*”.

3. Why using nanowire-shaped VN as the electrode? The authors used different characterizations on VN materials in manuscript. Thus the authors should give more explanation of this point in the Introduction part.

Response

We are grateful for the referee’s question. We selected nanowire-shaped VN as the electrode based on its following advantages: First, the large aspect ratio, intrinsic porous structure and nanometer sized dimension (<20 nm) of the VN nanowires endow them with high-rate capability and near surface Mg^{2+} intercalation pseudocapacitive behavior, which is the crucial factor for determining their superior rate capability and favorable cycling stability. Second, the nanowire shape microstructure could become interwoven with each other thereby avoiding aggregation upon cycling, which has been often seen when using nanoparticle or nanosheet morphologies.

Overall, the nanowire shape of the VN is key to achieving such good electrochemical performances of the VN electrode.

We have added the following discussion at **Lines 124-130** according to the referee's comment.

“It is anticipated that such nanowire-shaped VN is key to achieving favorable electrochemical performances when used as electrodes: 1) the large aspect ratio, intrinsic porous structure and nanometer sized dimension (<20 nm) of the VN nanowires endow them with high-rate capabilities and near surface Mg^{2+} intercalation pseudocapacitive behavior, which is the crucial factor for determining their superior rate capability and cycling stability. 2) the nanowire shape microstructure could become interwoven with each other thereby avoiding aggregation upon cycling, which has often been seen when using nanoparticle or nanosheet morphologies.”

4. Why the authors put Li^+ in the Faradaic reactions of VN in line 150? VN electrodes only show redox reaction in Mg^{2+} electrolyte. In Figure 2A, there are two oxidation peaks while only one reduction peak, can the authors explain this?

Response

Thanks for the referee's detailed comments on the electrochemical behavior of the VN electrode in the 1.0 M $MgSO_4$ electrolyte. In the original manuscript, we tended to discuss the Mg^{2+} intercalation pseudocapacitive mechanism of the VN electrode in line with a published work on the Li^+ intercalation VN system as a reference (Ref. 10: Lu et al., Nano Lett. 2013, 13, 2628-2633). Nevertheless, we now realize that this is quite misleading. Accordingly, we have revised the Faradaic reaction equation by removing the “ Li^+ ”.

To answer the second question on the reduction peak in Figure 2A, we re-tested the CV curve of VN in $MgSO_4$ at a scan rate of 10 mV s^{-1} and draw a comparison with that tested at a scan rate of 0.5 mV s^{-1} . In fact, there is a broad reduction peak for the CV curve tested at 10 mV s^{-1} , located at -0.4 V (vs. Ag/AgCl). This peak can be attributed to the pseudo-capacitive Mg ion intercalation/de-intercalation of the VN, which can be confirmed in the following part of the mechanistic study. As shown in Figure R3, the CV curve tested at 0.5 mV s^{-1} exhibits a similar two sets of oxidization and reduction peaks (marked by the arrows).

Figure R3. CV curves of VN in 1.0 M MgSO₄ tested at different scan rates (0.5 mV s⁻¹ and 10 mV s⁻¹).

5. Equivalent circuit fitting should be provided in Figure S2.

Response

Thanks for this suggestion. We have carried out the fitting following this comment.

The measured Nyquist plots were well fit based on an equivalent Randles circuit by using the following equation:

$$Z = R_s + \frac{1}{j\omega CPE + \frac{1}{R_{ct} + W_o}} + \frac{1}{j\omega C_l + \frac{1}{R_{leak}}}$$

where R_s is the internal resistance of the cell, CPE stands for the constant phase element (CPE), R_{ct} is the charge transfer resistance, W_o is the Warburg element, C_l is the low frequency mass capacitance, and R_{leak} is the low frequency leakage resistance.

For supercapacitors, especially the current system including pseudocapacitive processes, the fitted capacitive component in the EIS profile might not behave ideally, but act like a CPE instead. Regarding these EIS results for the VN electrodes in different aqueous cation systems, the Nyquist curves are better interpreted when CPE is replaced by the pure capacitance C_{dl} . The CPE component can be fit into two key parameters, *i.e.*, CPE-T and CPE-P according to the following equation:

$$Z_{CPE} = \frac{1}{T_0(j\omega)^n}$$

where CPE-T is related to the pseudocapacitance and CPE-P is related to the semi-circle in the Nyquist plot (the depressed semicircle), normally used for the notation ‘ n ’ in the equation. The more approximate to 1 the CPE-P parameter indicates the more ideal capacitive behavior of the electrochemical system.

Other parameters are also fit according to the non-linear calculation by defining different Randles circuit components. In the high frequency region, the point of intersection on the real axis represents the internal resistance R_s , which includes the intrinsic electronic resistance of the electrode material, the ohmic resistance of the electrolyte, and the interfacial resistance between the electrode and the current collector. The semicircle in the high frequency region provides the behavior of the interfacial charge transfer resistance R_{ct} . The Nyquist plot exhibits a straight long tail (that seems almost perpendicular to the x-axis) and stretches to the low frequency region. This line represents the mass capacitance C_l , and the inclined angle suggests a resistive element, which is the leakage resistance R_{leak} . The transmission line with an angle of nearly 45 degrees to the x-axis from high frequency to the mid-frequency represents the Warburg element W_o , which is expressed as:

$$W_o = \frac{A}{(j\omega)^n}$$

where A is the Warburg coefficient, ω is the angular frequency, and n is an exponent. All the value fittings using these elements are summarized in the following Table.

Figure R4. a) EIS spectra of VN electrode in different electrolytes. b) Detailed high-frequency region corresponding to (a). c) A Randles equivalent circuit. The equivalent circuit is used to fit the Nyquist profiles. d) Calculated b -value of the anodic peak and cathodic peak.

Table R2. Equivalent Circuit Parameters Obtained from the Fitting Results for Components of the Equivalent Circuit Fit with the Impedance Spectra

E_{chem} systems	R_s (Ω)	R_{ct} (Ω)	CPE-T (mF/cm^2)	CPE-P	A ($\Omega \cdot \text{S}^{-n}$)	n	C_F	R_{leak} (Ω)
VN in Li_2SO_4	4.22	0.79	3.04	0.90	1.02	0.43	0.75	0.82
VN in Na_2SO_4	6.92	1.56	1.28	0.84	1.16	0.45	0.25	0.51
VN in K_2SO_4	8.54	1.49	1.92	0.81	3.70	0.44	0.22	0.49
VN in MgSO_4	5.46	1.2	1.48	0.90	0.91	0.45	0.34	0.47

Following this comment, we have incorporated the relevant contents in the revised manuscript (**Supplementary Fig. 2; Supplementary Table 1; Supplementary Note 1**).

6. The authors should explain why the capacitive contribution part exceeds the total CV curve in Figure 2E.

Response

Thanks for this kind suggestion. In Figure 2E (the capacitive- and diffusion-controlled contribution curves), the capacitive contribution part exceeds the total CV curve in several places, which can be attributed to the fact that the reaction of the VN electrode in the Mg^{2+} ion electrolyte is not a completely capacitive response. The capacitive contributed CV curve derived from the tested CV results is an ideal calculation result, without considering the influence of the electrolyte ion dynamics resistance on the curve shape change or peak shift. However, for the real electrochemical pseudocapacitive process, the dynamic resistance can indeed lead to hysteresis in the electrochemical current response, which could result in an excursion from the predicted theoretical electrochemical behavior. Thus, this overflow curve is a normal phenomenon, which is determined by the difference between the real tested electrochemical current response and theoretically calculated results. Notably, there have also been published studies exhibiting the similar phenomenon regarding “capacitive contribution part exceeding the total CV curve”, as shown in Figure R5.

Figure R5. The capacitive contribution curves selected from different recent published papers. (a) Ref. 11: Wang *et al.*, *Adv. Mater.*, 2018, 30, 1800963, (b) Ref. 12: Liu *et al.*, *Adv. Energy Mater.*, 2019, 9, 1901379, (c) Ref. 13: Owusu *et al.*, *Nat. Commun.*, 2017, 8, 14264.

7. Although the authors conducted the operando XRD to prove the VN lattice shrinking during the Mg²⁺ intercalation, this phenomenon still needs more explanation.

Response

We are grateful to the referee for this suggestion. In our work, the reaction mechanism of VN in a neutral MgSO₄ electrolyte has been studied by *operando* XRD and *ex situ* XPS. The lattice shrinking of VN during the Mg²⁺ intercalation has been observed by the *operando* XRD tests. The possible reasons can be explained as follows:

(1) In comparison with other common monovalent cations (such as Li⁺, Na⁺, K⁺), Mg ions exhibit smaller hydrated ion size and stronger electrostatic interactions with their divalent nature. In an aqueous electrolyte system, the metal ions generally coordinate with multiple water molecules to form hydrated cations for subsequent electrochemical processes. The size of hydrated Mg²⁺ ion (Mg(H₂O)₆²⁺: ~2.10 Å) matches well with the lattice spacing of VN (002) lattice plane (~2.12 Å), but is smaller than those of other common hydrated ion systems, including Li(H₂O)₆⁺ (~2.15 Å), Na(H₂O)₆⁺ (~2.43 Å) and K(H₂O)₆⁺ (~2.80 Å). Therefore, the hydrated Mg²⁺ ions are not able to lead to a significant lattice expansion as compared to other larger-sized cations. In addition, the multi-valent nature of Mg²⁺ results in strong electrostatic interactions when intercalated into the lattice of the host VN electrode. (2) On the other hand, the crystal lattice shrinking can be also attributed to the small amount of intercalated ions. This has been clearly probed by a systematic study with respect to the Li-ion intercalation

pseudocapacitance of tungsten oxide characterized by *in situ* synchrotron X-ray probing and theoretical calculations *via* finite element analysis simulation and density functional theory calculation (Ref. 14: *Li et al., Nat. Commun., 2018, 9, 4798*). To sum up, the lattice shrinking during Mg^{2+} intercalation can be attributed to the strong electrostatic attraction of the divalent ions and the limited amounts of Mg ion intercalation, which is verified by the *ex situ* XPS results. The crystallite lattice shrinking has also been observed in other Mg^{2+} based intercalation systems (Ref. 15: *Song et al., Phys. Chem. Chem. Phys., 2015, 17, 5256-5264*; Ref. 16: *Tepavcevic et al., ACS Nano, 2015, 9, 8194-8205*) and other multi-ion intercalation systems (Ref. 17: Al^{3+} intercalation in $\text{W}_{18}\text{O}_{49}$: *Li et al., Small, 2017, 13, 1700380*).

8. Figure captions in Figure 3B-D are not correct.

Response

Thanks for spotting this error. We have corrected the order of the figure captions in **Figs. 3B-3D** in the revised manuscript.

9. The authors used VN and MnO₂ as negative and positive electrode, respectively. How did the authors balance the charges of these two electrodes in the quasi-solid-state ASCs and printed MASC?

Response

We thank the referee for this detailed question. The charge balance between the VN and MnO_2 electrodes is the key to optimizing the electrochemical performance of an asymmetric supercapacitor. This can be typically achieved by controlling the loading mass ratio of the active electrode materials. The mass ratio of the positive to negative electrode is theoretically calculated by adopting the following equation:

$$\frac{m^+}{m^-} = \frac{C^- \Delta V^-}{C^+ \Delta V^+}$$

where m^+ and m^- are the mass loading of the $\text{MnO}_2@C$ and VN electrodes, respectively. C^+ and ΔV^+ are the specific capacitance and the potential window of the $\text{MnO}_2@C$ electrode, respectively. C^- and ΔV^- are the specific capacitance and the potential window of the VN electrode, respectively.

Based on the three-electrode electrochemical measurements and the CV curve demonstrated in Figure 4B, the VN electrode exhibits a specific capacitance of 230 F g^{-1} with a potential window of 0.9 V, while the $\text{MnO}_2\text{@C}$ electrode shows a specific capacitance of 240 F g^{-1} with a broad potential window of 1.5 V. According to the loading mass balance calculation formula, the theoretical mass ratio of positive:negative electrode is therefore 1:1.74. To optimize the positive/negative electrode mass loading ratio, we have also tested the CV curves with different mass ratios as summarized in Figure R6. It is clear that the asymmetric supercapacitor with a mass ratio of 1.0:2.0 exhibits the widest stable working voltage of 2.2 V without any obvious gas-evolution-induced electrochemical polarization. As for the device with a mass ratio of 1.0:1.6, it also manifests an operating voltage of 2.2 V, but with the presence of an oxygen evolution reaction related peak. Herein it is worth mentioning that we are not able to achieve a precise control on the mass loading ratio between 1.0:1.6 and 1.0:2.0; hence, we decided to fabricate the quasi-solid-state ASC and printed MASC with a mass loading (mg cm^{-2}) ratio approaching $\sim 1.0:2.0$. For the screen printing process, we controlled the single printing of the $\text{MnO}_2\text{@C}$ positive electrode with a mass loading of 0.98 mg cm^{-2} and the VN negative electrode with a mass loading of 0.88 mg cm^{-2} (by comparing the mass difference before and after screen printing). Therefore, we simply screen printed the VN negative electrode twice and the $\text{MnO}_2\text{@C}$ positive electrode once for the final printed MASC device with a final mass loading ratio of 1.0:1.76 (the positive and negative electrode loading mass is 0.98 and 1.72 mg cm^{-2} , respectively).

Figure R6. CV curves with different mass ratios of positive and negative electrodes at a scan rate of 10 mV s^{-1} .

Based on this comment, we have incorporated the relevant contents into the revised manuscript (**Supplementary Fig. 22; Supplementary Note 2**).

10. It's unusual that the gel electrolyte-based system exhibited better rate performance and electrolyte ion diffusion behaviour than the liquid electrolyte-based system, as shown in Figure 4E-F. Why the intrinsic equivalent series resistance is larger in the PAM gel system but the Mg^{2+} diffusion is better?

Response

We are appreciative for this constructive comment. The PAM gel electrolyte is prepared by immersing a porous PAM film into a 1.0 M $MgSO_4$ electrolyte liquid. The advanced electrolyte ion diffusion behavior of the gel electrolyte as compared to the glass fiber separator based aqueous electrolyte system is mainly attributed to the porous structure, rich water content ratio of the PAM gel and the tight contact between the gel electrolyte and the electrode. These results have been cautiously repeated more than five times. We discuss the reasons in detail as follows:

First, the PAM gel exhibits a uniform and interconnected porous structure, which is more advanced than the glass fiber separator. Figure R7 shows the cross-sectional and planar SEM images of the PAM gel and the glass fiber after freeze-drying. It is evident that the PAM gel displays a micro-sized interconnected porous structure, which ensures the presence of abundant open channels for the electrolyte penetration and ion transport that is beneficial to ion diffusion.

Figure R7. **a)** Cross-section SEM images of the PAM gel (left) and **b)** glass fiber (right) respectively. **c)** Top view SEM images of the PAM gel (left) and **d)** glass fiber (right) respectively.

Second, the water content is an important parameter to characterize the capability of accommodating liquid electrolyte. The water content can be calculated by the equation below:

$$\eta = \frac{M_2 - M_1}{M_1}$$

where η is the water content (%), which represents the ability to fill the liquid aqueous electrolyte. M_1 is the mass before water absorption (g); M_2 is the mass after water absorption (g). Figure R8 summarizes the calculated water content results. Surprisingly, the water content of the PAM gel is 49% higher than that of the glass fiber separator. In addition, the intrinsic hydrophilic surface property of PAM also significantly increases the infiltration capability of the PAM gel framework. The interconnected porous structure and the hydrophilic surface properties appear to be the two major contributors to the higher water content ratio of the PAM gel electrolyte. A higher water content would not only ensure a more complete contact between the electrode and the electrolyte, but also slow down the evaporation of the electrolyte during subsequent processes thus improving the electrochemical performance (Ref. 18: *Li et al., Energy Environ. Sci., 2018, 11, 941-951*). In this sense, the feature of high water content may also be a reason for the better cycling stability for a long cycle test.

Figure R8. The water content comparison of a glass fiber (blue color) and a PAM gel (red color).

Finally, the perfect interfacial contact between the electrode materials and the PAM gel electrolyte significantly decreases the interfacial electrolyte ion transport resistance. Figure R9 shows a comparison between the real photos of asymmetric supercapacitors with PAM gel

electrolytes and glass fiber separators. Both negative and positive electrodes show quite tight contacts between the electrode and gel electrolyte. The electrode is difficult to strip off from the gel electrolyte. As a comparison, the electrodes can be easily peeled off from the glass fiber separator. To further characterize the interfacial contact between the electrode and the PAM gel electrolyte, cross-sectional SEM observation combined with energy-dispersive X-ray (EDS) elemental mapping of positive electrode//PAM gel//negative electrode full cell has been carried out. As shown in Figure R10a, there are tight interface contacts between the Ti mesh/VN or Ti mesh/MnO₂ electrode and the PAM gel electrolyte, where some electrode materials even embed themselves into the gel electrolyte. This tight contact endows the electrode with exceptional interfacial electrolyte transport and capacitive diffusion behavior of the electrolyte ions. The tight contact between the gel electrolyte/electrode interfaces are also verified by the cross-section EDX mapping analysis, as shown in Figure R10b.

Figure R9. a) Digital photos of the interface of a PAM gel electrolyte (top view and cross-sectional view). b) Digital photos of the interface of the glass fiber liquid electrolyte (top view and cross-sectional view).

Figure R10. a) SEM image of the interface between electrodes and a PAM gel. b) Elemental mapping of V, Mg and Mn at the interface.

The unique porous structure, high water content ratio, and tight gel electrolyte/electrode interfacial contact should all contribute to an enhanced ion diffusion, especially inside the PAM gel network and the gel electrolyte/electrode interface. EIS is an effective approach to characterize the charge transfer resistance and electrolyte ion diffusion behavior. Figure R11 further demonstrates the comparison of the EIS curves between the gel electrolyte and the glass fiber separator systems before and after the electrochemical tests. All these Nyquist plots have been fit according to the Randles equivalent circuit (summary in Table R3). R_s is the internal resistance, R_{ct} is the charge transfer resistance and n is the exponential factor in the open Warburg element. The value is close to 0.5, indicating the enhanced electrolyte ion diffusion behavior. As shown in Table R3, the R_s of the PAM gel electrolyte system only exhibits a negligible increase (0.037 Ω) after electrochemical measurements, which is smaller than the 0.243 Ω of the glass fiber separator system. The higher n value (0.41 and 0.43: before and after the test, respectively) of the gel electrolyte system as compared to 0.36 and 0.39 of the separator system indicates the better electrolyte ion diffusion. The much steeper line at the low frequency range in the Nyquist plots for the PAM gel system (especially after the electrochemical tests) also indicates enhanced ion diffusion. Such a better electrolyte ion diffusion capability is the key to achieving higher rate performance.

In summary, the better rate performance and better diffusion behavior of the PAM gel system can be attributed to the unique porous structure, high water content ratio and tight PAM gel electrolyte/electrode interfacial contact.

Figure R11. **a)** EIS spectra of the PAM gel electrolyte (red) and liquid electrolyte (purple) before cycle testing. **b)** The detailed high-frequency region corresponding to (a). **c)** EIS spectra of the PAM gel electrolyte (red) and liquid electrolyte (purple) after cycle testing. **d)** The detailed high-frequency region corresponding to (c). (Inserted image is a Randles equivalent circuit).

Table R3. Equivalent Circuit Parameters Obtained from the Fitting Results for Components of the Equivalent Circuit Fit with the Impedance Spectra

Electrolyte type	R_1 (Ω)	R_2 (Ω)	W_1 -P
PAM gel (before cycle)	1.981	0.852	0.406
PAM gel (after cycle)	2.018	0.895	0.434
Liquid (before cycle)	1.118	0.380	0.360
Liquid (after cycle)	1.361	0.785	0.396

Following this comment, we have incorporated the relevant contents into the revised manuscript (**Lines 267-278; Supplementary Figs. 8-12; Supplementary Table 2**).

“The superior rate performance presented in the gel electrolyte to that in the liquid electrolyte deserves further investigation. As such, comparative EIS profiles between the gel electrolyte and the liquid electrolyte systems before and after electrochemical tests have been collected (Fig. 4F; Supplementary Fig. 8). All these Nyquist plots have been fit according to a Randles equivalent circuit (Supplementary Table 2). It is evident that the R_s of the gel electrolyte system only exhibits a negligible increase (0.037 Ω) after electrochemical measurements, which is smaller than the 0.243 Ω of the liquid electrolyte system. The much steeper line at the low frequency range in the Nyquist plots for the gel system (especially after the electrochemical tests) also indicates enhanced ion diffusion. Such a better electrolyte ion diffusion capability is the key to achieving higher rate performance. Meanwhile, exhaustive morphological and elemental characterization provides evidence that the unique porous structure, high water content ratio and tight electrolyte/electrode interface contribute to the enhanced ion diffusion feature of the gel electrolyte-based system (Supplementary Figs. 9-12).”

11. The author should provide more details of electrochemical performance calculations in the experimental section.

Response

Thanks for this constructive advice. The calculations pertaining to the electrochemical performances are now described in detail in the **Method** section of the revised manuscript.

“The capacitance is calculated using the following equations:

According to the CV curves

$$C = \frac{1}{v * \Delta V} \int_{V_1}^{V_2} I(V) dV$$

In the three-electrode system, C is the capacitance (F), v is scan rate ($V s^{-1}$), I is the current response (A), $V_1(V_2)$ is the starting and ending voltage (V) and ΔV is the voltage window (V). For the single electrode, the specific capacitance is calculated according the following equation:

$$C_m = \frac{C}{2 * m}$$

where C_m is the mass specific capacitance ($F g^{-1}$), C is the capacitance (F) and m is the mass of active material (g; without the consideration of binders and conductive agents).

In the two-electrode system, the electrochemical performance is calculated according to the GCD curves

$$C = \frac{I * t}{\Delta V}$$

where C is the capacitance (F), I is the discharging current (A), t is the discharging time (s) and ΔV is the voltage window (V). For the device, the specific capacitance is calculated according the following equation:

$$C_m(s) = \frac{C}{m(s)}$$

where $C_m(s)$ is the mass or area specific capacitance ($F g^{-1}$ or $F cm^{-2}$), m is the total mass of positive and negative active material (g) and s is the area of positive or negative electrode (cm^{-2} , the area of positive and negative electrode is the same) and C is the capacitance (F).

The energy density (E) and power density (P) can be derived according to the following equations:

$$E_m = \frac{1}{2 * 3.6} C_m * V^2$$

where E_m is the mass energy density ($Wh kg^{-1}$), C_m is the mass specific capacitance ($F g^{-1}$) and V is the voltage (V).

$$E_v = \frac{1}{2 * 3600} C_v * V^2$$

where E_v is volume energy density ($Wh cm^{-3}$), C_v is the volume specific capacitance ($F cm^{-3}$, the volume is the product of electrode area and thickness, which includes the thickness of positive electrode, gel electrolyte and negative electrode) and V is the voltage (V).

As for the power density, the calculated equation is described as:

$$P = \frac{E}{t}$$

where P is the power density ($W kg^{-1}$ or $W cm^{-3}$), E is the energy density ($Wh kg^{-1}$ or $Wh cm^{-3}$) and t is the discharging time (h).”

12. The printed MASC which was used as the energy storage unit was integrated with PV in a solar-charging self-powered system, thus the self-discharge of MASC is an important parameter for evaluation of this self-powered system. What is the self-discharge performance of printed MASC?

Response

We appreciate the excellent comment about the self-discharging performance of a printed MASC. Indeed, it is necessary to evaluate the self-discharge behavior. For a supercapacitor, self-

discharge is one of the most significant challenges for practical applications. As shown in Figure R12, we have probed the self-discharge performance of the printed MASC by recording the open circuit voltage from the maximum 2.2 V to 1.0 V, which is similar to the standard self-discharge evaluation process demonstrated in a previous report (*M.F. El-Kady et al., Nat. Commun., 2013, 4, 1475*¹⁹). Our printed MASC self-discharges in 12 h to maintain 1.1 V (half the maximum voltage) and 18 h to maintain 1.0 V, outperforming most of the reported results (*Jiang et al., Nano Energy, 2018, 45, 266-272*²⁰ (0.6 to 0.4 V in 2 h); *Choi et al., Nat. Commun., 2016, 7, 13811*²¹ (1.0 to 0.8 V in 1.4 h); *Su et al., Energy Environ. Sci., 2014, 7, 2652-2659*²² (2.5 to 1.0 V in 8.3 h)).

Figure R12. The self-discharge curves of the MASCs.

Following this comment, we have incorporated the relevant contents in the revised manuscript (**Lines 315-316; Supplementary Fig. 15**).

13. The bending test of flexible integrated solar-charging self-powered units seems not enough for practical application. The capacitance retention reached 74 % after 20 cycles. Can authors improve the capacitance retention after bending test?

Response

Thanks for the referee's inspirational suggestion. We have improved the bending cycles of the self-powered units, which has updated in Figure R13. The self-powered units maintain 80% capacitance after 100 cycles of 90 degrees of repeated bending.

Figure R13. Durability test of a flexible solar-charging integrated unit for 100 cycles under 90 degrees bending cycles.

Following this comment, we have incorporated the relevant contents in the revised manuscript (**Lines 379-380; Fig. 6F**).

14. The line width of the interdigitated electrodes and the pitch between two fingers have great influence on the performances of the micro-asymmetric supercapacitors. Please give these values.

Response

We thank the reviewer for the constructive advice with regards to the details of the micro-pattern. In this work, two-step screen-printing technology has been employed. The size of the interdigitated electrodes is accordingly shown below (Figure R14). The line width is 1 mm, where the gap between two fingers is 1 mm and the gap between the two different electrodes at both ends is 0.6 mm.

Figure R14. Detailed dimensions of the screen-printing pattern.

Following this comment, we have incorporated the relevant contents in the revised manuscript (**Supplementary Fig. 14**).

15. *A plastic film was used to wrap the device but what is the plastic film. As the long-term stability (specific capacitance changing with time) is essential for the practical application of the micro-asymmetric supercapacitors, thus supplement the related tests.*

Response

Thanks for the detailed comment on the device sealing method. In fact, the plastic film is a PET transparent film and the lead connected position is tightly sealed by pasting AB gum, thereby avoiding leakage of electrolyte. Figure R15 displays the cyclic performance of the MASC at a current density of 2 mA cm^{-2} . Obviously, the MASC shows favorable long-term cycling stability. In the first 5000 cycles, the capacitance retention is maintained at 94%. Even after 8000 cycles, the capacitance retention could still maintain 90%, demonstrating good cycle stability.

Figure R15. Long-term cycling stability of a MASC at a charge/discharge current density of 2 mA cm^{-2} .

Following this comment, we have incorporated the relevant contents in the revised manuscript (**Lines 315-316; Supplementary Fig. 16**).

16. *The assembly of flexible solar-charging integrated units, especially the interconnect between the MASC and the Si solar cells is ambiguous. Please give detailed descriptions of the structure of this integrated system in the text and supplement the detailed fabrication process in the experimental section.*

Response

We fully agree with the referee that the detailed introduction of the integrated process should be described. Since the rear side of the flexible solar cell is uneven, directly printing the micro-supercapacitor on the rear of the solar cell is impractical. To tackle the non-smooth surface of the flexible solar cell, acrylic double-sided tape is pasted onto the backside of the solar cell first. Then, the cleaned PI substrate is attached to the other side of the acrylic tape, which is used for the printed substrate of the MASCs. The related fabrication process is illustrated in Figure R16.

Figure R16. The assembly process of the solar charging integrated system.

Following this comment, we have incorporated the relevant contents in the revised manuscript (**Lines 489-494; Supplementary Fig. 24**).

17. To catch broad interest, the background of printed supercapacitors and flexible supercapacitors should be discussed extensively. Some related papers are suggested for references, such as: *Chemical Society Reviews*, 2019, 48, 3229-3264; *Advanced Materials Technologies*, 2019, 1900196; *Advanced Materials*, 2016, 28, 5242-5248; etc.

Response

The suggested references have been added (**Refs. [30-32]**) to enrich the content.

- 30 Zhang, Y. Z. et al. Printed supercapacitors: materials, printing and applications. *Chem. Soc. Rev.* **48**, 3229-3264 (2019).
- 31 Li, D. et al. A simple strategy towards highly conductive silver nanowire inks for screen printed flexible transparent conductive films and wearable energy storage devices. *Adv. Mater. Technol.* **4**, 2698-2705 (2019).
- 32 Zhang, Y. Z. et al. A simple approach to boost capacitance: flexible supercapacitors based on manganese oxides@MOFs via chemically induced in situ self-transformation. *Adv. Mater.* **28**, 5242-5248 (2016).

Referee: 2***Referee 2 (Remarks to the Author):***

This manuscript (NCOMMS-19-19708) describes solar charging self-powered units using printable Mg-ion quasi-solid-state asymmetric supercapacitors. The idea of “self-powered & integrated devices” is a current hotspot and has good application prospects. The written English is commendably good, but there are still some deficiencies. In general, the manuscript could be accepted after major revision.

Response

We thank the referee for these valuable comments. Our response follows each comment.

1) *MnO₂ and VN possess different charge storage abilities and the mass ratio of the MnO₂ cathode to the VN anode will affect the specific capacitance and energy density of assembled MnO₂//MgSO₄-polyacrylamide//VN ASCs devices. The mass ratio is the most critical factor for asymmetric devices. Please provide more information about the influence of the mass ratio of the anode and the cathode on the electrochemical performance of the device. In addition, how to control the screen-printing process effectively?*

Response

We thank the referee for this detailed question. The charge balance between the VN and MnO₂ electrodes is the key to optimizing the electrochemical performance of an asymmetric supercapacitor. This can be typically achieved by controlling the loading mass ratio of the active electrode materials. The mass ratio of the positive to negative electrode is theoretically calculated by adopting the following equation:

$$\frac{m^+}{m^-} = \frac{C^- \Delta V^-}{C^+ \Delta V^+}$$

where m^+ and m^- are the mass loading of the MnO₂@C and VN electrodes, respectively. C^+ and ΔV^+ are the specific capacitance and the potential window of the MnO₂@C electrode, respectively. C^- and ΔV^- are the specific capacitance and the potential window of the VN electrode, respectively.

Based on three-electrode electrochemical measurements and the CV curve demonstrated in Figure 4B, the VN electrode exhibits a specific capacitance of 230 F g^{-1} with a potential window of 0.9 V, while the $\text{MnO}_2@\text{C}$ electrode shows a specific capacitance of 240 F g^{-1} with a broad potential window of 1.5 V. According to the loading mass balance calculation formula, the theoretical mass ratio of the positive:negative electrode is 1:1.74. To optimize the positive/negative electrode mass loading ratio, we have also tested the CV curves with different mass ratios as summarized in Figure R17. It is clear that the asymmetric supercapacitor with a mass ratio of 1.0:2.0 exhibits the widest stable working voltage of 2.2 V without any obvious gas-evolution-induced electrochemical polarization. As for the device with a mass ratio of 1.0:1.6, it also manifests an operating voltage of 2.2 V, but with the presence of an oxygen evolution reaction related peak. Herein it is worth mentioning that we are not able to achieve a precise control over a mass loading ratio between 1.0:1.6 and 1.0:2.0; hence, we decided to fabricate the quasi-solid-state ASC and printed MASC with a mass loading (mg cm^{-2}) ratio approaching $\sim 1.0:2.0$. For the screen printing process, we control the single printing of the $\text{MnO}_2@\text{C}$ positive electrode with a mass loading of 0.98 mg cm^{-2} and the VN negative electrode with a mass loading of 0.88 mg cm^{-2} (by comparing the mass difference before and after screen printing). Therefore, we simply screen print the VN negative electrode twice and the $\text{MnO}_2@\text{C}$ positive electrode once for the final printed MASC device with a final mass loading ratio of 1.0:1.76 (the positive and negative electrode loading mass is 0.98 and 1.72 mg cm^{-2} , respectively).

Figure R17. CV curves with different mass ratios of positive and negative electrodes at a scan rate of 10 mV s^{-1} .

Based on this comment, we have incorporated the relevant contents in the revised manuscript (**Supplementary Fig. 22; Supplementary Note 2**).

2) Encapsulating material is important for flexible electronics, what is the encapsulating film for the integrated unit? How to ensure that the encapsulating material is impervious to water and oxygen during long-term testing? Detailed description of the packaging process is necessary.

Response

Thanks for the detailed question about the encapsulating technique used in this work. The plastic film mentioned in the manuscript is PET transparent thermoplastic film. With respect to the micro-supercapacitor, the assembly process is illustrated in Figure R18. Moreover, for the solar charging self-powered unit, the encapsulating material must have good light transmission. Thus, the PET plastic film is used and the lead connected position is sealed by pasting AB gum thereby avoiding leakage of electrolyte. The packaging process for the self-powered unit is displayed in Figure R19. Also note that the electrolyte used for the assembled MASC is the neutral PAM-MgSO₄ gel aqueous electrolyte, which is stable when exposed to oxygen/air and water.

Figure R18. The packaging process of the printable MASC.

Figure R19. The packaging process of the flexible solar charging unit.

Based on this comment, we have incorporated the relevant contents in the revised manuscript (**Lines 476-495; Supplementary Figs. 23 & 24**).

3) *The authors used “crosslink with each other” to describe starch molecules and acrylamide molecules (page 10 line 252), this is a misunderstanding of the concept of crosslink. In fact, crosslinking only occurs among acrylamide molecules chains; for starch molecules and acrylamide molecules chains, physical entanglement and interpenetration between molecular chains are the correct mechanism.*

Response

We thank the referee for this correction. In this regard, the starch molecules do physically entangle and interpenetrate, which formed the network structure with acrylamide molecular chains. Therefore, the related description has been corrected according to the referee’s comment (**Lines 254-256**).

“As expected, starch molecules and acrylamide molecules physically entangle and interpenetrate with each other to form an ion conductive network with favorable water absorption ability and mechanical robustness, as demonstrated in Fig. 4A.”

4) The explanation of the better rate performance for the gel electrolyte compared with the liquid electrolyte-based system (Figure 4E) is somewhat far-fetched, more convincing explanation or experiments are necessary.

Response:

We are grateful to the referee for this comment. Therefore, we repeated this experiment more than five times. Objectively, the ion diffusion resistance of the PAM gel electrolyte is better than the glass fiber separator (with a liquid electrolyte) leading to the superior ion diffusion behavior, resulting in the better rate performance at large current densities. The unusual electrochemical performance is attributed to the following points:

First, the PAM gel exhibits a uniform and interconnected porous structure, which is more advanced than the glass fiber separator. Figure R20a, b, and d shows the cross-sectional and planar SEM images of the PAM gel and glass fiber after freeze-drying. It is evident that the PAM gel displays a micro-sized interconnected porous structure, which ensures the presence of abundant open channels for the electrolyte penetration and ions transport that is beneficial to ion diffusion.

Figure R20. **a)** Cross-section SEM images of the PAM gel (left) and **b)** glass fiber (right) respectively. **c)** Top view SEM images of the PAM gel (left) and **d)** glass fiber (right) respectively.

Second, the water content is an important parameter to characterize the capability of accommodating liquid electrolyte. The water content can be calculated by the equation below:

$$\eta = \frac{M_2 - M_1}{M_1}$$

where η is the water content (%), which represents the ability to maintain water. M_1 is the mass before water absorption (g); M_2 is the mass after water absorption (g). Figure R21 summarizes the calculated water content results. Surprisingly, the water content of the PAM gel is 49% higher than that of the glass fiber separator. In addition, the intrinsic hydrophilic surface properties of PAM also significantly increases the infiltration capability of the PAM gel framework. The interconnected porous structure and the hydrophilic surface properties are likely the two major contributors to the higher water content ratio of the PAM gel electrolyte. A higher water content would not only ensure a more complete contact between the electrode and the electrolyte, but also slow down the evaporation of the electrolyte during subsequent processes thus improving the electrochemical performance (Ref. 18: *Li et al., Energy Environ. Sci., 2018, 11, 941-951*). In this sense, the feature of high water content may also be a reason for the better cycling stability for a long-term cycle test.

Figure R21. The water content comparison of a glass fiber (blue color) and a PAM gel (red color).

Finally, the excellent interfacial contact between the electrode materials and the PAM gel electrolyte significantly decreases the interfacial electrolyte ion transport resistance. Figure R22 exhibits a comparison between the real photos of the asymmetric supercapacitors with the PAM gel electrolyte and the glass fiber separator. Both negative and positive electrodes show quite tight contacts between the electrode and gel electrolyte. The electrode is difficult to strip off from the gel electrolyte. As a comparison, the electrodes can be easily peeled off from the glass fiber separator. To further characterize the interfacial contact between the electrode and the PAM gel electrolyte, cross-sectional SEM observation combined with energy-dispersive X-ray (EDS) elemental mapping of the positive electrode//PAM gel//negative electrode full cell has been

carried out. As shown in Figure R23a, there are tight interfacial contacts between the Ti mesh/VN or Ti mesh/MnO₂ electrode and the PAM gel electrolyte, where some electrode materials even embed themselves into the gel electrolyte. This tight contact achieves excellent interfacial electrolyte transport and capacitive diffusion behavior of the electrolyte ions. The tight contact between the gel electrolyte/electrode interfaces is also verified by cross-sectional EDX mapping analysis, as shown in Figure R23b.

Figure R22. a) Digital photos of the interface of the PAM gel electrolyte (top view and cross-sectional view). b) Digital photos of the interface of the glass fiber liquid electrolyte (top view and cross-sectional view).

Figure R23. a) SEM image of the interface between the electrodes and the PAM gel. b) Elemental mapping of V, Mg and Mn at the interface.

The unique porous structure, high water content ratio, and tight gel electrolyte/electrode interfacial contact can contribute to the enhanced ion diffusion, especially inside the PAM gel network and the gel electrolyte/electrode interface. EIS is an effective approach to characterize the charge transfer resistance and the electrolyte ion diffusion behavior. Figure R24 further

demonstrates the comparison of EIS curves between the gel electrolyte and the glass fiber separator systems before and after the electrochemical tests. All these Nyquist plots have been fit according to a Randles equivalent circuit (summary in Table R3) where R_s is the internal resistance, R_{ct} is the charge transfer resistance and n is the exponential factor in the open Warburg element. The value is close to 0.5, indicating the better electrolyte ion diffusion behavior. As shown in Table R4, the R_s of the PAM gel electrolyte system only exhibits a negligible increase (0.037 Ω) after electrochemical measurements, which is smaller than the 0.243 Ω of the glass fiber separator system. The higher n value (0.41 and 0.43: before and after the tests, respectively) of the gel electrolyte system as compared to 0.36 and 0.39 of the separator system indicates the better electrolyte ion diffusion. The much steeper line at the low frequency range in the Nyquist plots for the PAM gel system (especially after the electrochemical tests) also indicates enhanced ion diffusion. Such a better electrolyte ion diffusion capability is the key to achieving higher rate performance.

In summary, the better rate performance and better diffusion behavior of the PAM gel system can be attributed to the unique porous structure, high water content ratio and tight PAM gel electrolyte/electrode interfacial contact.

Figure R24. **a)** EIS spectra of the PAM gel electrolyte (red) and liquid electrolyte (violet) before cycle testing. **b)** Detailed high-frequency region corresponding to (a). **c)** EIS spectra of the PAM gel electrolyte (red) and the liquid electrolyte (violet) after cycle testing. **d)** Detailed high-frequency region corresponding to (c). (Inset for (a) and (c) are Randles equivalent circuits).

Table R4. Equivalent Circuit Parameters Obtained from the Fitting Results for Components of the Equivalent Circuit Fit with the Impedance Spectra

Electrolyte type	R_1 (Ω)	R_2 (Ω)	W_1 -P
PAM gel (before cycle)	1.981	0.852	0.406
PAM gel (after cycle)	2.018	0.895	0.434
Liquid (before cycle)	1.118	0.380	0.360
Liquid (after cycle)	1.361	0.785	0.396

Following this comment, we have incorporated the relevant contents in the revised manuscript (**Lines 267-278; Supplementary Figs. 8-12; Supplementary Table 2**).

5) Why use “mF cm⁻²” instead of keeping up with “F g⁻¹” of a single electrode when evaluating electrochemical performance of coin cell quasi-solid-state ASCs? Comparing the electrochemical performances of the assembled devices with a single electrode can embody the importance of mass ratio.

Response

We thank the referee for the kind suggestions. In fact, the areal capacitance we demonstrated in Figure 4E is the specific capacitance of the full device, not for the single electrode.

We also thank the referee for suggesting testing the electrochemical performance of a single electrode to optimize the positive and negative electrode mass ratio. To determine the mass ratio between the positive and negative electrode of MnO₂@C//VN ASC, the gravimetric capacitance of both electrodes were tested in a three-electrode configuration as demonstrated in **Fig. 2D** (main text) and **Supplementary Fig. 5b**.

From the Figure R25, the specific capacitance of VN is 230 F g^{-1} with a potential window of 0.9 V. For the $\text{MnO}_2@\text{C}$ electrode, the specific capacitance is 240 F g^{-1} with a broad potential window of 1.5 V. Thus, the mass ratio is determined by the equation:

$$\frac{m^+}{m^-} = \frac{C^- \Delta V^-}{C^+ \Delta V^+}$$

where m^+ and m^- are the mass loading of the $\text{MnO}_2@\text{C}$ and VN electrodes, respectively, C^- and ΔV^- are the specific capacitance and potential window of the VN electrode, while C^+ and ΔV^+ are the specific capacitance potential window of the $\text{MnO}_2@\text{C}$ electrode. According to the formula, the optimal mass ratio of positive and negative electrode is 1:1.74.

Figure R25. a) Mass specific capacitance of VN negative electrode at different scan rates. b) Mass specific capacitance of $\text{MnO}_2@\text{C}$ positive electrode at different scan rates.

6) The title is about “flexible solar-charging integrated units”, but the manuscript falls behind the title. A large part of the text is about the characterization of anode materials VN, yet the description about “flexible self-powered” is not substantial, only simple bending test was mentioned, other flexible characterizations such as twisting, folding, knotting, etc. were not implemented. Some characterizations of the anode material could be put in the supplementary information, since we are more interested in “flexible self-powered” and its application in realistic situations.

Response

We appreciate the valuable suggestion on the layout of this manuscript and strongly agree with the referee that a large portion of the contents is related to the characterization of the VN anode materials, which makes the content of the flexible self-powered part not substantial. Following the referee’s comment, we have revised Figure 1 by putting some parts of the VN

characterization results into the supplementary information. More importantly, we have also supplemented additional measurements to demonstrate the flexible capability of the “flexible self-powered unit”.

As for the concern about the excessive contents of VN, we have removed Figure 1E, F and G in the original main text and put them into supplementary information as Fig. S1. Although VN is not a novel material for supercapacitor applications, it is indeed a new material in the Mg^{2+} ion electrolyte system. It is therefore anticipated that this might attract wide interest from the readers of *Nature Communications*.

As for the complementary flexibility characterizations for flexible self-powered units: First, as demonstrated in Figure R26, after optimizing the fabrication process, the flexibility of a solar-charging integrated unit has achieved 80% capacitance retention after 90° bending over 100 cycles. Second, as shown in Figure R27, according to the referee’s suggestion, we have supplemented the solar charging and MASC discharging measurements under different mechanical deformations, such as folding, twisting and waving. The integrated self-powered units exhibit favorable flexible capability under a variety of deformation conditions, as illustrated by the photographs and detailed parameters from the top view and the side view in Figure R27b. We have measured the photo-charging and MASC discharge curves before and after folding, rolling and waving deformations (Figure R27c). The integrated units exhibit retention of 97.5%, 85.2% and 93.0%, respectively, indicative of great robustness.

Taken together, the solar charging self-powered unit exhibits favorable bending performance (80% after 100th bending at 90°) and work perfectly even after various structural deformations (bending, folding, rolling and waving). After these revisions, this work now puts more focus on the “flexible self-powered” integrated unit and its applications under realistic situations.

Figure R26. Durability test of a flexible solar-charging integrated unit for 100 cycles under 90 degrees bending recycle.

Figure R27. a) Solar-charging and electric discharging curves under different flexible conditions (folding, rolling and waving). b) The digital photos and detailed parameter of the integrated unit in three different flexible characterizations. c) Solar charging and electric discharging curves (initial state, after folding, after rolling and after waving)

Following this comment, we have incorporated the relevant contents in the revised manuscript **(Line 378-384; Fig. 6F; Supplementary Fig. 21)**.

“A mechanical durability test was also carried out by repeatedly bending the integrated device at 90° for 100 cycles (Fig. 6F and inset). The capacitance retention reached 94% and 80% after 10 and 100 bending cycles, respectively. In further contexts, various flexible conditions were implemented to verify the mechanical robustness of our devices (Supplementary Fig. 21). Before and after folding, rolling and waving deformations,

the integrated unit exhibited capacitance retentions of 97.5%, 85.2% and 93.0%, respectively. All these experiments provide strong evidence for the good flexibility of our self-powered unit for practical applications.”

7) Other errors, such as extra space (page 15 line 409), inappropriate molecular formula (Supplementary Figure 1a), also exist. Please double check the manuscript.

Response

We are very grateful to the referee for pointing out these errors. The mentioned errors have been corrected. Moreover, the whole manuscript has been thoroughly checked.

Referee: 3***Referee 3 (Remarks to the Author)***

In this report, Tian et al. reported the Mg-ion induced microsupercapacitors with self-charging solar-energy storage units for portable electronic applications. The on-chip microsupercapacitors were fabricated using vanadium nitride and manganese oxide-based materials, which demonstrated an energy density of 279 13.1 mWh cm⁻³ at a power density of 72 mW cm⁻³ with good cycling stability. Following, a flexible solar cell was assembled with microsupercapacitor as self-charging units, which demonstrated a maximum energy conversion/storage efficiency of 11.95%. It would be interesting finding, if the integrated device was assembled in the same housing by developing designs where electrodes are shared between the energy storage unit and the solar cell. In the current work, the authors have designed an externally connected microsupercapacitor with solar cell, which creates external ohmic voltage loss and leakage current. Considering the significance level of the prepared materials and the reported approach in the fabrication of microsupercapacitors and self-charging units are not new, as this kind of approaches have already been reported earlier for various portable applications, such as Wang et al, npj 2D Materials and Applications, 2, 7, 2018, Etienne et al., Electrochem. Commun. 28, 104, 2013, Maher et al, Proc. Natl. Acad. Sci., 2015, 112, 4233, Gao et al, Nat. Comm., 7, 11586, 2016, Yun et al. Nano Energy, 49 (2018), 644, Ahmad et al, Nano Lett. 2018, 18, 1856, Kim et al, J. Mater. Chem. A, 2017, 5, 1906). Therefore, I do not recommend this manuscript to be suitable for publication in a well-reputed journal like Nature Communications.

Response

We are grateful to the referee for their viewpoint and feedback.

First, we agree with the referee that the “same housing” design is a great idea and could serve as the main target for a solar-charging self-powered energy storage unit. In terms of the designs with electrodes shared by the energy storage unit and the energy harvesting unit (e.g., the solar cell), to the best of our knowledge, it is still a big challenge to harvest a high energy storage efficiency or energy conversion efficiency from the shared electrode without sacrificing the dual function. As summarized in a recent perspective paper (*Gurung et al. Joule, 2018, 2, 1217–1230*²³), the highest overall efficiency of an integrated system with shared electrode design reaches only 7.61% by combining the Si solar cell with a Li₄Ti₅O₁₂/LiCoO₂ Li-ion battery via

sharing an Al current collector (please see Table R5: A large portion of the “same-housing” device delivers an overall efficiency lower than 5%). **In our current discrete solar cell-asymmetric supercapacitor design, we have managed to achieve an overall efficiency of 17.57%, which is likely a new record in the field of a solar charging energy storage system.** The discrete design is beneficial to optimizing the energy storage unit and solar cell to match the maximum power point of the PV module with the charging voltage and current of the energy storage module, without worrying about the balance in the shared electrode performance between the photoelectric and electrochemical processes. Considering these aspects, the “externally connected design” could still be a feasible solution, at least at the current stage, to achieve a high overall efficiency solar-charging energy storage integrated system. Indeed, we agree with the referee that “the same housing” design is of great significance to the practicability of these devices and thus needs to be advanced. Ongoing studies in our labs deal with the practical design of a “same housing” integrated system targeting high performance in realistic circumstances.

Table R5. Related Parameters of the Solar Charging Integrated System in the Same Housing Unit

System	Integrated type	$\eta_{(\text{overall})}$	Output voltage	Cycle number	Ref.
PSC//LIB	Common electrode	7.61 %	5.40 V	100 (98 %)	24
DSC//SSC	Common electrode	1.50 %	0.60 V	N/A	25
PSC//SSC	Common electrode	1.10 %	0.70 V	N/A	26
PSC//SSC	Common electrode	5.26 %	0.84 V	6	27
PSC//SSC	Common electrode	4.70 %	0.71 V	50	28
PPB	Same electrode	0.034 %	2.90 V	10 (<50 %)	29
DSC//LIB	Common electrode	0.82 %	3.00 V	3	30
PFB	Photo electrode	3.00 %	1.20 V	N/A	31
PFB	Photo electrode	0.11 %	0.90 V	10	32
PFB	Photo electrode	1.25 %	0.43 V	20	33
PFB	Photo electrode	0.71 %	1.50 V	20 (<71 %)	34

Annotation: PSC: perovskite solar cell; DSC: dye sensitized solar cell; SSC: symmetric supercapacitor; LIB: Li ion battery; PPB: photo-rechargeable perovskite battery; PFB: photo-charging flow battery.

Second, we thank the referee for the feedback on the novelty of this work. We apologize for our carelessness on not fully demonstrating the novelty in the original manuscript. Indeed, the

novelty is not located in the material choices of electrodes or the fabrication method of the micro-supercapacitor. The significance of this work is mainly embodied in three aspects: **i) a first-time demonstration and study on Mg-ion intercalation pseudocapacitive behavior of the VN electrode *via operando* characterizations; ii) a new record of overall efficiency at 17.57 % for an integrated solar-charging energy storage system; and iii) an integration of a printed flexible and wearable MnO₂@C//VN micro-chip asymmetric supercapacitor and a solar cell with outstanding self-charging performance and flexibility.** These concrete significance advances are discussed in detail as follows:

1. VN is a commonly used negative electrode material for an asymmetric supercapacitor. However, its electrochemical performance, such as the specific capacitance and stable working voltage, are limited by the employed electrolyte (basic KOH or monovalent neutral electrolyte, LiCl or Na₂SO₄). Herein, for the first time we report the multivalent ion (i.e. Mg²⁺) intercalation pseudocapacitive behavior of VN with outstanding specific capacitance and stable working potential. As demonstrated in Figure R28 (Figure 2 in the main text of the manuscript), the Mg²⁺ intercalated VN system exhibits markedly enhanced specific capacitance and a larger stable potential as compared to those of other neutral monovalent cation systems. Simultaneously, the Mg²⁺ intercalation pseudocapacitive charge storage behavior has also been studied by *ex situ* XPS and *in situ* XRD in detail, as shown in Figure R29 (Figure 3 in the main text of the manuscript). Along these lines, an innovative charge storage mechanism has been accordingly proposed on a basis of highly reversible Mg²⁺ intercalation/de-intercalation behavior during the charging/discharging process.

Figure R28. a) Gravimetric specific capacitances in different electrolytes at a scan rate of 10 mV s^{-1} . b) Polarization curves of a VN electrode in different cation-based neutral electrolytes.

Figure R29. Electrochemical reaction mechanism of VN in a neutral MgSO_4 electrolyte. a) Operando XRD patterns of VN during the charge/discharge process (current density: 0.1 A g^{-1}). b) Schematic diagram showing the contraction and expansion of the (200) crystal plane during the charge/discharge process. c) *Ex situ* XPS Mg 1s spectra at different potentials during the CV scan at a scan rate of 5 mV s^{-1} . d) Change of vanadium valence state according to XPS V 2p spectra, which corresponds to (b).

- Upon the combination of a GaAs solar cell with the $\text{MnO}_2@\text{C}/\text{VN}$ Mg-ion asymmetric supercapacitor, the overall efficiency of the unit harvests, to the best of our knowledge, a new record of 17.57% with one sun illumination and discharge current of 0.2 mA (0.1 A g^{-1}). As demonstrated in Figure R30, with the light intensity increasing from 1000 W m^{-2} to 1500 W m^{-2} , it is interesting to find that the overall efficiency can be further improved to 18.3% from 17.57%, which is attributed to the excellent performance under the large sunlight intensity of the GaAs solar cell. Even altering the discharge current from 0.2 to 1.0 mA, the overall efficiency can still be maintained at $>15.52\%$. Despite being combined with a relatively low solar conversion efficiency (15.46%) enabled amorphous Si solar cell, our integrated solar-charging unit can still harvest an overall efficiency of

11.95%, which outperforms most of the solar charging self-powered systems reported thus far.

Figure R30. a) Response photocurrent at different light intensities of a GaAs solar cell. b) Voltage-time profiles of our solar-charging integrated units with different discharge currents under 1000 W m^{-2} . c) Voltage-time profiles of our solar-charging integrated units with different discharge currents under 1500 W m^{-2} . d, e) The overall efficiency calculated according to b and c at different discharge currents. f) J-V curve of the GaAs solar cell.

3. The high overall efficiency is attributed to the good match between the input charging voltage of the GaAs solar cell with the asymmetric supercapacitor, the great tolerance of the fluctuating current by the asymmetric supercapacitor, the high solar conversion efficiency of the GaAs solar cell and the favorable energy storage efficiency. The relative parameters are summarized in Table R6 (all reported studies included herein are based on the discrete design between a solar cell and an energy storage module). To efficiently charge the energy storage module of a solar charging integrated unit, the open circuit voltage (V_{oc}) should be higher than the maximum working voltage of the energy storage device. The V_{oc} of the GaAs solar cell is 2.4 V, while the V_{max} is 2.2 V for the $\text{MnO}_2@C//\text{VN}$ asymmetric supercapacitor. Thus, it offers a good match for the solar conversion and energy storage modules. In addition, the large current fluctuations and high short circuit current (usually hundreds of mA for solar cells) could destroy a battery-type energy storage device. Our asymmetric supercapacitor exhibits excellent tolerance of current fluctuations and large input currents. As demonstrated in the I-V curve in

Figure R30f, the GaAs solar cell displays an extremely high solar conversion efficiency of 25.88%. The energy storage efficiency (η_{storage}) of the energy storage device is an important parameter that has a crucial influence on the overall efficiency. Based on the unique electrochemical performance of our asymmetric supercapacitor, the thus-derived integrated system can harvest an energy storage efficiency of 67.9%.

Table R6. Related Parameters of the Rigid Solar Charging Self-powered System with a Discrete Design

Conversion system	Storage system	Overall efficiency	Conversion efficiency	Storage efficiency	Output voltage (V)	Endurance current (max)	Ref.
GaAs	ASC	17.57 %	25.88 %	67.90 %	2.20	114 mA cm ⁻²	This work
p-Si	ASC	11.95 %	15.46 %	77.31 %	2.20	114 mA cm ⁻²	work
GaAs	FB	14.10 %	26.10 %	54.02 %	1.25	50 mA cm ⁻²	35
PSC	LIB	9.36 %	14.40 %	69.44 %	3.14	4 C	36
DSC	LIB	5.50 %	7.89 %	69.71 %	3.14	4 C	36
PSC	LIC	8.41 %	15.00 %	56.07 %	3.00	5 A g ⁻¹	37
PSC	LIB	7.80 %	12.65 %	61.66 %	2.60	2 C	38
PSC	SSC	10.00 %	13.60 %	73.53 %	1.45	15 mA cm ⁻²	39
PSC	SSC	5.10 %	6.10 %	83.61 %	1.20	N/A	40
OSC	SSC	2.92 %	5.20 %	56.15 %	0.53	10 mA cm ⁻²	41

Annotation: PSC: perovskite solar cell; DSC: dye sensitized solar cell; p-Si: polysilicon; c-Si: crystalline silicon; OSC: organic solar battery; ASC: asymmetric supercapacitor; SSC: symmetric supercapacitor; FB: flow battery; LIB: Li ion battery; LIC: Li ion capacitor.

- The Mg-ion asymmetric supercapacitor can be simply screen-printed and combined with the solar cell in a scalable fashion. As summarized in Table R6, our integrated system exhibits a favorable long-term cycling stability accompanied by good device flexibility. It can retain 98.7% of its initial capacitance after 100 bending cycles and manifests a capacitance retention of 97.46% even upon deformation tests such as folding, rolling and waving. Moreover, after 100 bending cycles at 90 degrees, the capacitance retention is still maintained at 80%. The related performances of other reported flexible solar charging self-powered units are summarized in Table R7. As for flexible integrated systems, our integrated unit demonstrates the highest overall efficiency and output voltage amongst the state-of-the-art systems, which can be attributed to the in-plane structural design and employment of an asymmetric supercapacitor with a high voltage

window. In general, the printing technology provides a feasible method to fabricate self-powered integrated units, which possess high efficiency and flexibility.

Table R7. Related Parameters of the Flexible Solar Charging Self-powered System

System	Structure	$\eta_{(\text{overall})}$	Output voltage	Cycle number	Ref.
a-Si//MASC	In plane	5.20 %	2.20 V	100 (98.70 %)	This work
DSC//SSC	In plane	N/A	1.10 V	10 (54.55 %)	42
DSC//SSC	Fiber	0.90 %	1.20 V	NA	43
DSC//SSC	In plane	N/A	1.80 V	10	44
SC//ASC	Fiber	N/A	1.55 V	NA	45
a-Si//SSC	In plane	N/A	0.85 V	NA	46
DSC//SSC	Fiber	1.80 %	0.70 V	5	47
PSC//ASC	In plane	4.90 %	0.60 V	10	48
PC//SSC	Fiber	0.85 %	0.60 V	1000	49

Annotation: PSC: perovskite solar cell; DSC: dye sensitized solar cell; a-Si: amorphous silicon; PC: polymer solar battery; MASC: micro-asymmetric supercapacitor; ASC: asymmetric supercapacitor; SSC: symmetric supercapacitor.

Finally, we thank the referee for the listed references for comparison on a basis of their mastery of knowledge in this field. Upon carefully reviewing the seven pioneer reports listed, we conclude that we are not able to pinpoint any similarity or novelty we have just addressed above. In detail, Ref. (*Wang et al. npj 2D Materials and Applications 2018, 2, 7* and *Etienne et al. Electrochem. Commun. 2013, 28, 104*) reports on the electrochemical performances of VN as the negative electrode in two specific asymmetric supercapacitors in a KOH aqueous electrolyte. The maximum working voltages are respectively 1.5 V and 1.8 V, which does not jeopardize the importance of our current work on the Mg-ion neutral electrolyte system. Ref. (*Maher et al, Proc. Natl. Acad. Sci. 2015, 112, 4233* and *Yun et al. Nano Energy 2018, 49, 644*) simply demonstrate the solar cell charging of micro-supercapacitors, without systematically discussing the overall efficiency or storage efficiency. Ref. (*Gao et al, Nat. Commun, 2016, 7, 11586*) studied the output voltage of a solar cell. However, their study only exhibited an energy storage efficiency of ~43%, without showing the overall efficiency. Ref. (*Ahmad et al, Nano Lett. 2018, 18, 1856*) demonstrated an interesting system by combining a lead halide perovskite solar cell with a Li-ion battery to build a “photo-battery”. Nevertheless, it can only harvest an overall efficiency of 0.034%. Ref. (*Kim et al, J. Mater. Chem. A 2017, 5, 1906*) assembled the best

performance solar charging integrated unit (with a max overall efficiency of 10.97%) amongst all the listed references, but it is still far behind the performance of our current system (17.57%).

Taken together, in the present work, we have studied the Mg-ion intercalation pseudocapacitive behavior of the VN electrode for the first time and achieved the high capacitance of the electrode. A new record of overall efficiency 17.57% of an integrated solar-charging energy storage system under 1 sun has been realized, which can be attributed to the optimized electrode/electrolyte material, the favorable choice of each module and the integrated device design. We use printing technology to enable the smooth combination of micro-supercapacitors with solar cells to build-up a fully flexible solar charging self-powered unit, thereby offering versatile chances for future applications in realistic scenarios. Accordingly, the statement of novelty has been reflected in our revision (**Fig. 6C; Supplementary Fig. 19; Supplementary Tables 4 & 5, as well as the discussion therein**).

Supporting references

- 1 Ma, X. *et al.* High Energy Density Micro-Supercapacitor Based on a Three-Dimensional Bicontinuous Porous Carbon with Interconnected Hierarchical Pores. *ACS Appl. Mater. Interfaces* **11**, 948-956 (2019).
- 2 Qiu, M., Sun, P., Cui, G., Tong, Y. & Mai, W. A Flexible Microsupercapacitor with Integral Photocatalytic Fuel Cell for Self-Charging. *ACS Nano* **13**, 8246-8255 (2019).
- 3 Xie, J.-Q. *et al.* In situ growth of Cu(OH)₂@FeOOH nanotube arrays on catalytically deposited Cu current collector patterns for high-performance flexible in-plane micro-sized energy storage devices. *Energy Environ. Sci.* **12**, 194-205 (2019).
- 4 Kim, S.-W. *et al.* Plotter-assisted integration of wearable all-solid-state micro-supercapacitors. *Nano Energy* **50**, 410-416 (2018).
- 5 Gao, J. *et al.* Laser-Assisted Large-Scale Fabrication of All-Solid-State Asymmetrical Micro-Supercapacitor Array. *Small* **14**, 1801809 (2018).
- 6 Yang, W. *et al.* Carbon-MEMS-Based Alternating Stacked MoS₂@rGO-CNT Micro-Supercapacitor with High Capacitance and Energy Density. *Small* **13**, 1700639 (2017).
- 7 Zhang, L. *et al.* Flexible Micro-Supercapacitor Based on Graphene with 3D Structure. *Small* **13**, 1603114 (2017).
- 8 Zhang, C. J. *et al.* Additive-free MXene inks and direct printing of micro-supercapacitors. *Nat. Commun.* **10**, 1795 (2019).
- 9 Guo, R. *et al.* In-Plane Micro-Supercapacitors for an Integrated Device on One Piece of Paper. *Adv. Funct. Mater.* **27**, 1702394 (2017).
- 10 Lu, X. *et al.* High energy density asymmetric quasi-solid-state supercapacitor based on porous vanadium nitride nanowire anode. *Nano Lett.* **13**, 2628-2633 (2013).
- 11 Wang, X. *et al.* Caging Nb₂O₅ Nanowires in PECVD-Derived Graphene Capsules toward Bendable Sodium-Ion Hybrid Supercapacitors. *Adv. Mater.* **30**, 1800963 (2018).
- 12 Liu, Y. *et al.* A Large Scalable and Low-Cost Sulfur/Nitrogen Dual-Doped Hard Carbon as the Negative Electrode Material for High-Performance Potassium-Ion Batteries. *Adv. Energy Mater.* **8**, 1901379 (2019).
- 13 Owusu, K. A. *et al.* Low-crystalline iron oxide hydroxide nanoparticle anode for high-performance supercapacitors. *Nat. Commun.* **8**, 14264 (2017).
- 14 Li, K. *et al.* Lattice-contraction triggered synchronous electrochromic actuator. *Nat. Commun.* **9**, 4798 (2018).
- 15 Song, J. *et al.* Activation of a MnO₂ cathode by water-stimulated Mg(2+) insertion for a magnesium ion battery. *Phys. Chem. Chem. Phys.* **17**, 5256-5264 (2015).
- 16 Sanja Tepavcevic. *et al.* Nanostructured Layered Cathode for Rechargeable Mg-Ion Batteries. *ACS Nano* **9**, 8194-8205 (2015).
- 17 Li, K. *et al.* Aluminum-Ion-Intercalation Supercapacitors with Ultrahigh Areal Capacitance and Highly Enhanced Cycling Stability: Power Supply for Flexible Electrochromic Devices. *Small* **13**, 1700380 (2017).

- 18 Li, H. *et al.* An extremely safe and wearable solid-state zinc ion battery based on a hierarchical structured polymer electrolyte. *Energy Environ. Sci.* **11**, 941-951 (2018).
- 19 El-Kady. *et al.* Scalable fabrication of high-power graphene micro-supercapacitors for flexible and on-chip energy storage. *Nat. Commun.* **4**, 1475 (2013).
- 20 Jiang, Z. *et al.* Ultrahigh-Working-Frequency Embedded Supercapacitors with 1T Phase MoSe₂ Nanosheets for System-in-Package Application. *Adv. Funct. Mater.* **29**, 1807116 (2018).
- 21 Choi, C. *et al.* Improvement of system capacitance via weavable superelastic bicrolled yarn supercapacitors. *Nat. Commun.* **7**, 13811 (2016).
- 22 Su, Z. *et al.* Scalable fabrication of MnO₂ nanostructure deposited on free-standing Ni nanocone arrays for ultrathin, flexible, high-performance micro-supercapacitor. *Energy Environ. Sci.* **7**, 2652-2659 (2014).
- 23 Gurung, A. *et al.* Solar Charging Batteries: Advances, Challenges, and Opportunities. *Joule* **2**, 1217-1230 (2018).
- 24 Um, H.-D. *et al.* Monolithically integrated, photo-rechargeable portable power sources based on miniaturized Si solar cells and printed solid-state lithium-ion batteries. *Energy Environ. Sci.* **10**, 931-940 (2017).
- 25 Liu, R. *et al.* Integrated solar capacitors for energy conversion and storage. *Nano Research* **10**, 1545-1559 (2017).
- 26 Liu, R. *et al.* A photocapacitor based on organometal halide perovskite and PANI/CNT composites integrated using a CNT bridge. *J. Mater. Chem. A* **5**, 23078-23084 (2017).
- 27 Liu, Z. *et al.* Novel Integration of Perovskite Solar Cell and Supercapacitor Based on Carbon Electrode for Hybridizing Energy Conversion and Storage. *ACS Appl. Mater. Interfaces* **9**, 22361-22368 (2017).
- 28 Xu, J. *et al.* Integrated Photo-Supercapacitor Based on PEDOT Modified Printable Perovskite Solar Cell. *Adv. Mater. Technologies* **1**, 1600074 (2016).
- 29 Ahmad, S. *et al.* Photo-Rechargeable Organo-Halide Perovskite Batteries. *Nano Lett.* **18**, 1856-1862 (2018).
- 30 Guo, W. *et al.* An integrated power pack of dye-sensitized solar cell and Li battery based on double-sided TiO₂ nanotube arrays. *Nano Lett.* **12**, 2520-2523 (2012).
- 31 Cheng, Q. *et al.* Photorechargeable High Voltage Redox Battery Enabled by Ta₃N₅ and GaN/Si Dual-Photoelectrode. *Adv. Mater.* **29**, 1700312 (2017).
- 32 Tian, Z. *et al.* Solar-driven capacity enhancement of aqueous redox batteries with a vertically oriented tin disulfide array as both the photo-cathode and battery-anode. *Chem. Commun.* **55**, 1291-1294 (2019).
- 33 Zhou, Y. *et al.* Efficient Solar Energy Harvesting and Storage through a Robust Photocatalyst Driving Reversible Redox Reactions. *Adv. Mater.* **30**, 1802294 (2018).
- 34 Lei, B. *et al.* A solar rechargeable battery based on hydrogen storage mechanism in dual-phase electrolyte. *Nano Energy* **38**, 257-262 (2017).

- 35 Li, W. *et al.* 14.1% Efficient Monolithically Integrated Solar Flow Battery. *Chem* **4**, 2644-2657 (2018).
- 36 Gurung, A. *et al.* Highly Efficient Perovskite Solar Cell Photocharging of Lithium Ion Battery Using DC-DC Booster. *Adv. Energy Mater.* **7**, 1602105 (2017).
- 37 Li, C. *et al.* Flexible perovskite solar cell-driven photo-rechargeable lithium-ion capacitor for self-powered wearable strain sensors. *Nano Energy* **60**, 247-256 (2019).
- 38 Xu, J. *et al.* Efficiently photo-charging lithium-ion battery by perovskite solar cell. *Nat. Commun.* **6**, 8103 (2015).
- 39 Xu, X. *et al.* A Power Pack Based on Organometallic Perovskite Solar Cell and Supercapacitor. *ACS Nano* **9**, 1782-1787 (2015).
- 40 Liang, J. *et al.* An all-inorganic perovskite solar capacitor for efficient and stable spontaneous photocharging. *Nano Energy* **52**, 239-245 (2018).
- 41 Lechêne, B. P. *et al.* Organic solar cells and fully printed super-capacitors optimized for indoor light energy harvesting. *Nano Energy* **26**, 631-640 (2016).
- 42 Cai, J. *et al.* A. High-performance all-solid-state flexible carbon/TiO₂ micro-supercapacitors with photo-rechargeable capability. *RSC Adv.* **7**, 415-422 (2017).
- 43 Chai, Z. *et al.* Tailorable and Wearable Textile Devices for Solar Energy Harvesting and Simultaneous Storage. *ACS Nano* **10**, 9201-9207 (2016).
- 44 Dong, P. *et al.* A flexible solar cell/supercapacitor integrated energy device. *Nano Energy* **42**, 181-186 (2017).
- 45 Gao, Z. *et al.* Cotton-textile-enabled flexible self-sustaining power packs via roll-to-roll fabrication. *Nat. Commun.* **7**, 11586 (2016).
- 46 Manjakkal, L. *et al.* Flexible self-charging supercapacitor based on graphene-Ag-3D graphene foam electrodes. *Nano Energy* **51**, 604-612 (2018).
- 47 Liang, J. *et al.* MoS₂-Based All-Purpose Fibrous Electrode and Self-Powering Energy Fiber for Efficient Energy Harvesting and Storage. *Adv. Energy Mater.* **7**, 1601208 (2017).
- 48 Zhang, F. *et al.* Highly flexible and scalable photo-rechargeable power unit based on symmetrical nanotube arrays. *Nano Energy* **46**, 168-175 (2018).
- 49 Zhang, Z. *et al.* Integrated polymer solar cell and electrochemical supercapacitor in a flexible and stable fiber format. *Adv. Mater.* **26**, 466-470 (2014).

REVIEWERS' COMMENTS:

Reviewer #1 (Remarks to the Author):

My concerns have been addressed and the manuscript has been much improved. I would like to recommend acceptance.

Reviewer #2 (Remarks to the Author):

The authors have satisfactorily answered all the questions and carefully revised the manuscript, therefore, I suggest to publish this work.